# Generative Modeling of Irregular Time Series via SDE-Induced Continuous-Discrete Variational Inference

**Zexin Yuan** [1]  **Qinliang Su** [1 2]  **Junxi Xiao** [1]

## Abstract

Neural Stochastic Differential Equations(SDEs) are widely adopted for modeling irregular time series which is ubiquitous in the real world. We introduce SDE-VI, a novel variational inference framework for generative modeling of irregular time series that proposes a paradigm shift from learning posterior SDE over the whole continuous time interval directly to the SDE-induced joint distribution over discrete-time observations. Specifically, we directly learn a variational posterior and ensure it is induced by Linear Time-Varying SDEs, as a stable and scalable inference backbone. SDE-VI learns a variational posterior that is guaranteed to be induced by a Linear Time-Varying (LTV) SDE, providing a stable and scalable inference backbone. To overcome the long-standing trade-off between computational efficiency and expressivity in prior work, we further generalize the framework to nonlinear, complex-valued SDE-induced variational inference, enabling intricate dynamics modeling for real-world data. Extensive experiments across healthcare, physics, climate, and IoT benchmarks demonstrate state-of-the-art performance on interpolation, extrapolation, regression, and classification tasks.

## 1. Introduction

Time-series data is ubiquitous across domains ranging from finance and healthcare to the natural sciences, underpinning critical real-world applications such as financial risk assessment, healthcare monitoring and weather forecasting. Most models assume all observations are collected at regular intervals and that all variables are measured at the same

time points. However, real-world data often violate these assumptions due to stochastic factors like sensor clock drift or transmission latency, as well as intrinsically irregular events in trading (Turkmen et al., 2019), metrology (Menne et al., 2016) and physiology (Goldberger et al., 2000). These realistic challenges have motivated extensive research on irregular time series over the last decade (Li & Marlin, 2015; Lipton et al., 2016; Futoma et al., 2017; Che et al., 2018; Rubanova et al., 2019; Shukla & Marlin, 2021).

Recently, Neural Continuous-Discrete State Space Models (CD-SSMs) (Chen et al., 2018; Li et al., 2020; Kidger et al., 2020; Park et al., 2025; Ansari et al., 2023; Course & Nair, 2023; Zeng et al., 2023; Schirmer et al., 2022; Shukla & Marlin, 2021; Oh et al., 2024) have gained increasing popularity for modeling the continuous dynamics of irregular time series. Specifically, such models assume the *discrete* observations are generated from a latent *continuous* process governed by neural differential equations. Despite their strong expressiveness, existing Neural CD-SSMs suffer from critical limitations. Ordinary differential equation (ODE)-based models are fundamentally limited to the assumption of deterministic system trajectories, thus failing to capture complex systems characterized by inherent stochasticity (e.g., financial markets) and multimodality (e.g., autonomous driving) (Liao et al., 2025; Salzmann et al., 2020). In contrast, Stochastic Differential Equation (SDE)-based models excel at capturing complex dynamics and uncertainty. However, effectively learning these SDE-based models remains a significant challenge. Some approaches(Schirmer et al., 2022; Ansari et al., 2023) employ Kalman-filtering-based algorithms (Kalman, 1960), which is restricted to simple Linear Gaussian State Space Models (LGSSM) and rely on recursive updating. Other methods resort to variational inference with approximate posterior SDEs. However, standard approaches (Li et al., 2020; Zeng et al., 2023) are often computationally prohibitive and unstable (Course & Nair, 2023). To improve stability and scalability, recent works (Course & Nair, 2023; Park et al., 2025) compromise by restricting the approximate posterior to linear SDEs with strict decaying constraints, facing severe expressivity bottleneck. Additionally, these methods induce non-Markovian posteriors, yielding distorted trajectories around observation points(Ansari et al., 2023; Park

---

[1]School of Computer Science and Engineering, Sun Yat-sen University, Guangzhou, China [2]Guangdong Key Laboratory of Big Data Analysis and Processing, Guangzhou, China. Correspondence to: Qinliang Su <suqliang@mail.sysu.edu.cn>.

*Proceedings of the 43rd International Conference on Machine Learning*, Seoul, South Korea. PMLR 306, 2026. Copyright 2026 by the author(s).

et al., 2025).

To overcome these limitations, we propose a paradigm shift. While existing methods endeavor to explicitly characterize the posterior over the continuous time interval, we recognize that observations are available only at irregular, discrete timestamps. Thus, we argue that instead of directly inferring the path measure over the continuous-time interval, it suffices to focus on the learning of the joint distribution over these discrete timestamps. However, the difficulty is that we have to ensure the joint distribution over the discrete timestamps is strictly governed by an underlying SDE, thereby conforming to the underlying structure in SDEs.

Based on this insight, we propose SDE-VI, a novel variational inference framework for the generative modeling of irregular time series. First, we directly learn a variational posterior distribution that is guaranteed to be induced by a valid Linear Time-Varying (LTV) SDE. While this linear backbone ensures stability and scalability, it inherently lacks the capacity to capture nonlinear and multimodal dynamics in the real world. To bridge this gap, we generalize our framework to nonlinear-SDE-induced variational inference. Specifically, we employ Conditional Normalizing Flows to refine the variational posterior distribution and prove that the refined distribution is induced by a nonlinear SDE. Furthermore, we generalize our variational inference framework to the complex domain to explicitly incorporate inductive biases for modeling oscillatory dynamics (periodic or quasi-periodic, e.g. pendulum), which are critical for modeling real-world time series. Extensive experiments on four benchmarks across healthcare, physics, climate and IoT domains demonstrate that SDE-VI achieves state-of-the-art performance in interpolation, extrapolation, regression and classification tasks.

## 2. Related Works

**Continuous-time RNNs, Transformers** Standard RNNs (Cho et al., 2014; Hochreiter & Schmidhuber, 1997) and Transformers (Vaswani et al., 2017) lack mechanisms to capture continuous dynamics. To bridge this gap, typical solutions incorporate heuristic decay, such as exponentially decaying hidden states (Che et al., 2016; Cao et al., 2018) or modifying attention weights to penalize elapsed time (Zhang et al., 2019). Alternatively, Shukla & Marlin (2021) employs periodic time embeddings within a cross-attention mechanism to map irregular inputs to a regular latent space. These methods are limited to assumptions of specific decaying or periodic patterns.

**Neural ODEs** Chen et al. (2018) introduced Neural ODEs to model continuous dynamics via vector fields parameterized by neural networks. To incorporate online observations, Rubanova et al. (2019) and Lechner & Hasani

(2020) sequentially update hidden states at observation times, whereas Kidger et al. (2020) leverages rough path theory to process continuous data streams. De Brouwer et al. (2019) and Herrera et al. (2021) introduce specific mechanisms for instantaneous state jumps. These approaches rely on numerical solvers and are fundamentally limited in modeling stochasticity and multimodal trajectories.

**Neural SDEs** One line of research focuses on learning through approximate posterior SDEs. Li et al. (2020) proposes a variational evidence lower bound and uses the stochastic adjoint sensitive method to compute gradients. Zeng et al. (2023) focuses on stochastic dynamics in homogeneous latent spaces. However, employing non-linear approximate SDEs mandates expensive sequential simulation and suffers from stiff gradients (Course & Nair, 2023). To improve stability and scalability, Course & Nair (2023) restricts the variational posterior to linear and Park et al. (2025) restricts both posterior and prior to locally linear time-invariant SDEs. While scalable, linear SDEs lack expressivity for complex dynamics. Moreover, the ad-hoc parameterization of the posterior drift renders the latent process non-Markovian(Park et al., 2025; Ansari et al., 2023). Another category extends Deep Kalman Filters to continuous time. Schirmer et al. (2022) employs locally linear priors, whereas Ansari et al. (2023) utilizes Jacobian-based linearizations of non-linear priors. Nevertheless, these methods remain constrained by the Linear Gaussian State Space Model (LGSSM) and rely on recursive Kalman updating.

## 3. Preliminaries

We consider a multivariate time series dataset $\mathcal{X} = \{\mathbf{x}_{t_i}\}_{i=0}^N$ of irregularly sampled at discrete timestamps $0 = t_0 < \cdots < t_N = T$. To model the underlying continuous-time dynamics, a predominant approach (Li et al., 2020; Zeng et al., 2023; Park et al., 2025; Course & Nair, 2023; Ansari et al., 2023) is to assume $\mathcal{X}$ are generated from a latent stochastic process $\mathbf{z}_{[0,T]} = \{\mathbf{z}_t\}_{t \in [0,T]}$ governed by an Itô Stochastic Differential Equation (SDE):

$$d\mathbf{z}_t = \mathbf{f}_\theta(t, \mathbf{z}_t)dt + \mathbf{L}_\theta(t, \mathbf{z}_t)d\mathbf{W}_t, \quad \mathbf{z}_0 \sim p(\mathbf{z}_0), \quad (1)$$

where $\mathbf{f}_\theta : [0,T] \times \mathbb{R}^D \to \mathbb{R}^D$ is the drift function; $\mathbf{L}_\theta : [0,T] \times \mathbb{R}^D \to \mathbb{R}^{D \times D}$ is the diffusion coefficient; $\mathbf{W}_t$ is a standard $D$-dimensional Wiener process; and $p(\mathbf{z}_0)$ is the distribution of the initial state $\mathbf{z}_0$. Each discrete observation $\mathbf{x}_{t_i}$ is assumed to be conditionally independent given the latent state $\mathbf{z}_{t_i}$. Therefore, the generative process can be represented as:

$$d\mathbb{P}_\theta(\mathcal{X}, \mathbf{z}_{[0,T]}) = d\mathbb{P}_\theta(\mathbf{z}_{[0,T]}) \times \left[ \prod_{i=0}^N p_\theta(\mathbf{x}_{t_i}|\mathbf{z}_{t_i})d\mathbf{x}_{t_i} \right],$$
$$(2)$$

where $\mathbb{P}_\theta(\mathbf{z}_{[0,T]})$ denotes the path measure induced by the SDE in (1) and the emission model $p_\theta(\mathbf{x}|\mathbf{z})$ is typically a likelihood function like $\mathcal{N}(\mathbf{x}; \mathbf{e}_\theta(\mathbf{z}), \sigma^2\mathbf{I})$, where $\mathbf{e}_\theta(\mathbf{z})$ denotes a neural network.

Our objective is to maximize the marginal log-likelihood $\log p_\theta(\mathcal{X}) = \log \int \left[\prod_{i=0}^N p_\theta(\mathbf{x}_{t_i}|\mathbf{z}_{t_i})\right] d\mathbb{P}_\theta(\mathbf{z}_{[0,T]})$. Due to intractability of the integral in calculating $p_\theta(\mathcal{X})$, a predominant paradigm (Li et al., 2020; Zeng et al., 2023; Course & Nair, 2023; Park et al., 2025) is to maximize the path-based Evidence Lower Bound (ELBO):

$$\mathcal{L}_{\text{path}}(\boldsymbol{\theta}, \boldsymbol{\phi}) = \mathbb{E}_{\mathbb{Q}_\phi}\left[\sum_{i=0}^N \log p_\theta(\mathbf{x}_{t_i}|\mathbf{z}_{t_i})\right] - D_{\text{KL}}(\mathbb{Q}_\phi || \mathbb{P}_\theta)$$
(3)

where $\mathbb{Q}_\phi$ denotes the path measure induced by a SDE:

$$d\mathbf{z}_t = \mathbf{g}_\phi(t, \mathbf{z}_t)dt + \mathbf{H}_\phi(t, \mathbf{z}_t)d\mathbf{W}_t, \quad \mathbf{z}_0 \sim q_\phi(\mathbf{z}_0). \quad (4)$$

To ensure the path-based KL divergence $D_{KL}(\mathbb{Q}_\phi || \mathbb{P}_\theta)$ well-defined, the diffusion function $\mathbf{H}_\phi(t, \mathbf{z}_t)$ is required to be identical to the prior one $\mathbf{L}_\theta(t, \mathbf{z}_t)$. It can be easily seen that $\log p_\theta = \mathcal{L}_{\text{path}}$ holds if and only if $D_{\text{KL}}(\mathbb{Q}_\phi || \mathbb{P}^*) = 0$, where $\mathbb{P}^* = \mathbb{P}_\theta(\mathbf{z}_{[0,T]}|\mathcal{X})$ is the exact posterior path measure, which is induced by the true posterior SDE (Park et al., 2025):

$$d\mathbf{z}_t = [\mathbf{f}_\theta(t, \mathbf{z}_t) + \mathbf{L}_\theta\mathbf{L}_\theta^\top\nabla_\mathbf{z}\log h(t, \mathbf{z}_t)]dt + \mathbf{L}_\theta(t, \mathbf{z}_t)d\mathbf{W}_t,$$
(5)

where $h(t, \mathbf{z}_t) = \mathbb{E}_{\mathbb{P}_\theta}[\prod_{j:t_j>t} p(\mathbf{x}_{t_j}|\mathbf{z}_{t_j})|\mathbf{z}_t]$ represents the expected future likelihood. By Girsanov Theorem, $D_{\text{KL}}(\mathbb{Q}_\phi || \mathbb{P}^*) = 0$, if and only if $\mathbf{g}_\phi(t, \mathbf{z}_t) = \mathbf{f}_\theta(t, \mathbf{z}_t) + \mathbf{L}_\theta\mathbf{L}_\theta^\top\nabla_\mathbf{z}\log h(t, \mathbf{z}_t)$ almost everywhere. However, since the emission likelihood $p(\mathbf{x}|\mathbf{z})$ is typically parameterized by non-linear neural networks, the term $\nabla_\mathbf{z}\log h$ is analytically intractable. To approximate $\mathbf{f}_\theta(t, \mathbf{z}_t) + \mathbf{L}_\theta\mathbf{L}_\theta^\top\nabla_\mathbf{z}\log h(t, \mathbf{z}_t)$, existing works (Li et al., 2020; Zeng et al., 2023; Park et al., 2025) propose to use a neural networks conditioned on a time-varying context vector $\mathbf{c}_t$ to model the drift coefficient as $\mathbf{g}_\phi(t, \mathbf{z}, \mathbf{c}_t)$, which, however, renders the latent process $\mathbf{z}_{[0,T]}$ non-Markovian due to the inclusion of future information in the context vector $\mathbf{c}_t$. Moreover, to approximate the path-based ELBO $\mathcal{L}_{\text{path}}$, expensive numerical simulations over the continuous-time stochastic path conforming to $\mathbb{Q}_\phi$ is required. To avoid such simulations, recent methods (Course & Nair, 2023; Park et al., 2025) restrict the variational SDE to linear forms such that $\mathbb{Q}_\phi$ can be efficiently and analytically solved:

$$d\mathbf{z}_t = [\mathbf{F}_\phi(t)\mathbf{z}_t + \mathbf{b}_\phi(t)]dt + \mathbf{L}_\phi(t)d\mathbf{W}_t, \quad \mathbf{z}_0 \sim q_\psi(\mathbf{z}_0),$$
(6)

where the drift function $\mathbf{g}_\phi(t, \mathbf{z}_t)$ is set to a linear function of $\mathbf{z}_t$ as $\mathbf{g}_\phi(t, \mathbf{z}_t) = \mathbf{F}_\phi(t)\mathbf{z}_t + \mathbf{b}_\phi(t)$; $q_\psi = \mathcal{N}(\boldsymbol{\mu}_0, \boldsymbol{\Sigma}_0)$, $\mathbf{F}_\phi$, $\mathbf{L}_\phi$ and $\boldsymbol{\Sigma}_0$ are all set as diagonal matrices. As mentioned above, $D_{\text{KL}}(\mathbb{Q}_\phi || \mathbb{P}_\theta)$ is well-defined only if $\mathbf{L}_\phi = \mathbf{L}_\theta$. To

ensure closed-form solvability of the SDE, the prior diffusion coefficient $\mathbf{L}_\theta(t, \mathbf{z})$ is set as a state-independent diagonal matrix $\mathbf{L}_\theta(t)$, limiting its expressivity for modeling stochasticity. To further improve computational efficiency, Park et al. (2025) set $\mathbf{F}_\phi, \mathbf{b}_\phi$ to be constant between two adjacent observations, keep $\mathbf{L}_\phi$ constant across all observations, and set $\mathbf{L}_\theta = \mathbf{L}_\phi$, $\mathbf{f}_\theta = \mathbf{F}_\phi\mathbf{z}_t$. From the discussion above, we can see that the existing linear SDE-based methods limit the diffusion coefficient to the same state-independent form for both prior and posterior, and force the drift coefficient $\mathbf{f}_\theta$ in the prior SDE to adapt to an overly simplified posterior structure, making it difficult to model complex nonlinear dynamics in the real world.

# 4. Method

## 4.1. Learning Posterior SDE via Marginal-based ELBO

While existing methods endeavor to explicitly characterize the posterior over the continuous time interval $[0, T]$, we notice that we only have observations at irregular discrete timestamps. Thus, we argue that instead of directly inferring the path measure over the continuous-time interval, we only need to focus on the learning of the joint distribution over these discrete timestamps. Specifically, by letting $\mathcal{Z} = \{\mathbf{z}_{t_i}\}_{i=0}^N$, We denote $p_\theta(\mathcal{Z})$ and $q_\phi(\mathcal{Z}|\mathcal{X})$ as the joint prior and variational posterior distributions over these discrete-time states, governed by the dynamics of the prior SDE in (1) and the variational Linear Time-Varying (LTV) SDE in (6), respectively. We refer to such joint distributions arising from continuous-time SDE as *SDE-induced distributions*. We can then formalize our objective as the marginal-based ELBO:

$$\mathcal{L}_{\text{marg}}(\boldsymbol{\theta}, \boldsymbol{\phi}) = \mathbb{E}_{q_\phi}\left[\sum_{i=0}^N \log p_\theta(\mathbf{x}_{t_i}|\mathbf{z}_{t_i})\right] - D_{\text{KL}}(q_\phi || p_\theta).$$
(7)

We can prove that $\mathcal{L}_{\text{marg}}$ is a superior objective than the path-based ELBO $\mathcal{L}_{path}$, as stated below:

**Theorem 4.1.** *Given any approximate posterior SDE, its inducing continuous-time measure $\mathbb{Q}$ and discrete-time joint distribution $q$, the following inequality holds:*

$$\mathcal{L}_{marg}(q) \geq \mathcal{L}_{path}(\mathbb{Q}). \quad (8)$$

Note that evaluating $\mathcal{L}_{\text{marg}}$ does not require the path-based KL divergence $D_{\text{KL}}(\mathbb{Q}_\phi || \mathbb{P}_\theta)$ to be well-defined. Hence, we can let the prior diffusion $\mathbf{L}_\theta(t, \mathbf{z})$ independent from the variational term $\mathbf{L}_\phi(t)$, thereby enhancing the stochastic expressivity of the prior model.

It is known that that $q_\phi$ induced by the SDE in (6) maintains

a Gauss-Markov distribution (Arnold, 1974) structure:

$$q_\phi(\mathcal{Z}) = q_\phi(\mathbf{z}_{t_0}) \prod_{i=0}^{N-1} q_\phi(\mathbf{z}_{t_{i+1}} | \mathbf{z}_{t_i}), \qquad (9)$$

where we have omitted the symbol $\mathcal{X}$ for notation conciseness. The conditional marginal $q_\phi(\mathbf{z}_{t_{i+1}} | \mathbf{z}_{t_i})$ is determined by the pairwise marginal $q_\phi(\mathbf{z}_{t_i,t_{i+1}})$, which is a joint Gaussian:

$$q_\phi(\mathbf{z}_{t_i,t_{i+1}}) = \mathcal{N}\left(\begin{bmatrix} \mathbf{z}_{t_i} \\ \mathbf{z}_{t_{i+1}} \end{bmatrix}; \begin{bmatrix} \boldsymbol{\mu}_{t_i} \\ \boldsymbol{\mu}_{t_{i+1}} \end{bmatrix}, \begin{bmatrix} \boldsymbol{\Sigma}_{t_i} & \boldsymbol{\Sigma}_{t_i,t_{i+1}} \\ \boldsymbol{\Sigma}_{t_i,t_{i+1}}^\top & \boldsymbol{\Sigma}_{t_{i+1}} \end{bmatrix}\right), \qquad (10)$$

Since $\mathbf{F}_\phi$, $\mathbf{L}_\phi$ and $\boldsymbol{\Sigma}_0$ in (6) are all assumed to be diagonal, covariance matrix $\boldsymbol{\Sigma}_t$ is also diagonal, that is, $\boldsymbol{\Sigma}_t = \mathrm{diag}(\boldsymbol{\sigma}_t^2)$, and cross-covariance $\boldsymbol{\Sigma}_{t_i,t_{i+1}}$:

$$\boldsymbol{\Sigma}_{t_i,t_{i+1}} = \mathrm{diag}(\boldsymbol{\sigma}_{t_i} \odot \boldsymbol{\gamma}_{t_i,t_{i+1}} \odot \boldsymbol{\sigma}_{t_{i+1}}). \qquad (11)$$

where $\odot$ denotes Hadamard product and $\boldsymbol{\gamma}_{t_i,t_{i+1}}$ means the correlation coefficient between $\mathbf{z}_{t_i}$ and $\mathbf{z}_{t_{i+1}}$ in a dimension-wise manner.

The variational posterior distribution $q_\phi(\mathcal{Z})$ is induced by the LTV-SDE in (6). Thus, the distribution parameters, $\{\boldsymbol{\mu}_{t_i}, \boldsymbol{\sigma}_{t_i}, \boldsymbol{\gamma}_{t_i,t_{i+1}}\}_{i=0}^N$ is a function of the SDE parameters $\{\mathbf{F}_\phi, \mathbf{b}_\phi, \mathbf{L}_\phi\}$. Specifically, the continuous moments $\{\boldsymbol{\mu}_t, \boldsymbol{\Sigma}_t\}_t$ can be solved using the Lyapunov differential equations (Särkkä & Solin, 2019), yielding the results:

$$\boldsymbol{\mu}_{t_i} = e^{\int_0^{t_i} \mathbf{F}_\phi(s)ds} \boldsymbol{\mu}_0 + \int_0^{t_i} e^{\int_\tau^{t_i} \mathbf{F}_\phi(s)ds} \mathbf{b}_\phi(\tau)d\tau,$$

$$\boldsymbol{\Sigma}_{t_i} = e^{2\int_0^{t_i} \mathbf{F}_\phi(s)ds} \boldsymbol{\Sigma}_0 \qquad (12)$$

$$+ \int_0^{t_i} e^{2\int_\tau^{t_i} \mathbf{F}_\phi(s)ds} \left(\mathbf{L}_\phi(\tau)\mathbf{L}_\phi(\tau)^\top\right)d\tau.$$

For any adjacent $t_i$ and $t_{i+1}$, the correlation coefficient $\boldsymbol{\gamma}_{t_i,t_{i+1}}$ (see Appendix A.2 for details):

$$\boldsymbol{\gamma}_{t_i,t_{i+1}} = \exp\left(\int_{t_i}^{t_{i+1}} \mathbf{F}_\phi(s)ds\right) \boldsymbol{\sigma}_{t_i} \oslash \boldsymbol{\sigma}_{t_{i+1}}, \qquad (13)$$

where $\oslash$ denotes Hadamard division.

According to (12) and (13), $\{\boldsymbol{\mu}_{t_i}, \boldsymbol{\sigma}_{t_i}, \boldsymbol{\gamma}_{t_i,t_{i+1}}\}_{i=0}^N$ is determined by $\{\mathbf{F}_\phi, \mathbf{b}_\phi, \mathbf{L}_\phi\}$. Therefore, $\{\mathbf{F}_\phi, \mathbf{b}_\phi, \mathbf{L}_\phi\}$ can be learned by maximizing $\mathcal{L}_{\mathrm{marg}}$ in theory. However, directly optimizing $\{\mathbf{F}_\phi, \mathbf{b}_\phi, \mathbf{L}_\phi\}$ via $\mathcal{L}_{\mathrm{marg}}$ presents severe numerical challenges. As shown in (12), the state moments $\boldsymbol{\mu}_{t_i}$ and $\boldsymbol{\Sigma}_{t_i}$ depend on the cumulative integral $\exp\left(\int_0^{t_i} \mathbf{F}_\phi(s)ds\right)$. Over long time horizons, this term becomes exponentially sensitive to $\mathbf{F}_\phi$, leading to exploding gradients and numerical instability in practice. Consequently, prior works (Course & Nair, 2023; Park et al., 2025) constrain $\mathbf{F}_\phi \prec \mathbf{0}$ to ensure training stability. However, this constraint forces the system to forget information exponentially fast, thereby severely limiting its ability to capture long-term dependencies.

## 4.2. LTV-SDE-Induced Variational Inference

We observe that the objective function $\mathcal{L}_{\mathrm{marg}}$ depends directly on the distribution parameters $\{\boldsymbol{\mu}_{t_i}, \boldsymbol{\sigma}_{t_i}, \boldsymbol{\gamma}_{t_i,t_{i+1}}\}_{i=0}^N$, rather than the SDE parameters $\{\mathbf{F}_\phi, \mathbf{b}_\phi, \mathbf{L}_\phi\}$. Therefore, instead of learning the posterior SDE parameters $\{\mathbf{F}_\phi, \mathbf{b}_\phi, \mathbf{L}_\phi\}$, we propose to directly optimize the distribution parameters $\{\boldsymbol{\mu}_{t_i}, \boldsymbol{\sigma}_{t_i}, \boldsymbol{\gamma}_{t_i,t_{i+1}}\}_{i=0}^N$. However, not every arbitrary set of distribution parameters $\{\boldsymbol{\mu}_{t_i}, \boldsymbol{\sigma}_{t_i}, \boldsymbol{\gamma}_{t_i,t_{i+1}}\}_{i=0}^N$ corresponds to a valid underlying SDE; in other words, the learned parameters might not be inducible by any SDE. To ensure that the learned parameters $\{\boldsymbol{\mu}_{t_i}, \boldsymbol{\sigma}_{t_i}, \boldsymbol{\gamma}_{t_i,t_{i+1}}\}_{i=0}^N$ indeed correspond to a valid SDE, we prove that they must satisfy the following conditions:

**Theorem 4.2** (Existence of a Valid SDE). *The discrete Gauss-Markov distribution $q_\phi(\mathcal{Z})$ parameterized by $\{\boldsymbol{\mu}_{t_i}, \boldsymbol{\sigma}_{t_i}, \boldsymbol{\gamma}_{t_i,t_{i+1}}\}_{i=0}^N$ is induced by a valid diagonal LTV-SDE if and only if the following conditions hold: 1) $\boldsymbol{\mu}(t)$ and $\boldsymbol{\sigma}(t)$ are continuously differentiable functions with $\boldsymbol{\sigma}(t) \succ \mathbf{0}$; 2) for any adjacent $t_i, t_{i+1}$ with interval $\Delta t_{i,i+1} \triangleq t_{i+1} - t_i$, the correlation coefficient $\boldsymbol{\gamma}_{t_i,t_{i+1}}$ can be written as:*

$$\boldsymbol{\gamma}_{t_i,t_{i+1}} = \exp\left(-\bar{\mathbf{s}}(t_i, t_{i+1}) \cdot \Delta t_{i,i+1}\right), \qquad (14)$$

*where $\bar{\mathbf{s}}(t_i, t_{i+1}) \succ \mathbf{0}$ is a bounded continuous function.*

The constraint on $\boldsymbol{\gamma}_{t_i,t_{i+1}}$ aligns well with our physical intuition: it ensures $\boldsymbol{\gamma}_{t_i,t_{i+1}} \to \mathbf{1}$ as $\Delta t_{i,i+1} \to 0$ and $\boldsymbol{\gamma}_{t_i,t_{i+1}} \to \mathbf{0}$ as $\Delta t_{i,i+1} \to \infty$. Moreover, we can prove $\bar{\mathbf{s}}$ corresponds to the average rate of entropy increment in a phase space. Hence, we term this as *Entropy Increment Parameterization(EIP)*. Details are provided in Appendix A.4.

For a set of distribution parameters $\{\boldsymbol{\mu}_{t_i}, \boldsymbol{\sigma}_{t_i}, \boldsymbol{\gamma}_{t_i,t_{i+1}}\}_{i=0}^N$ satisfying the above conditions, we show that it can be induced by lots of Linear Time-Varying (LTV) SDEs. However, under the piece-wise Linear Time-Invariant (LTI) constraint, the underlying SDE is unique. Please refer to Appendix A.3.

Since the posterior parameters $\{\boldsymbol{\mu}_{t_i}, \boldsymbol{\sigma}_{t_i}, \boldsymbol{\gamma}_{t_i,t_{i+1}}\}_{i=0}^N$ depend on observation $\mathcal{X}$, in practice, we parameterize $\{\boldsymbol{\mu}_{t_i}, \boldsymbol{\sigma}_{t_i}, \boldsymbol{\gamma}_{t_i,t_{i+1}}\}_{i=0}^N$ as outputs of neural networks with $\mathcal{X}$ as input. To ensure that the learned parameters correspond to a valid SDE, we design the network architecture according to the constraints derived in Theorem 4.2. Specifically, observations $\mathcal{X}$ are first projected into input embeddings and then fed into a lightweight Transformer (Vaswani et al., 2017) encoder to capture contextual information. Furthermore, to explicitly capture continuous-time dynamics, we project the time information into high-dimensional continuous embeddings. Following prior work (Song et al., 2021; Rombach et al., 2022; Park et al., 2025), we employ sinusoidal Fourier embeddings to embed timestamps

(Vaswani et al., 2017) and add the time embeddings to the input of each Transformer block. As a result, the transformer encoder outputs context-rich embeddings $\{\mathbf{h}_{t_i}\}_{i=0}^{N}$ that encapsulate both observations information and arbitrary timestamps. Finally, we parameterize the estimators $\mathbf{f}_{\boldsymbol{\mu}}, \mathbf{f}_{\boldsymbol{\sigma}}, \mathbf{f}_{\boldsymbol{\gamma}}$ by applying shallow heads on top of the embeddings $\{\mathbf{h}_{t_i}\}_{i=0}^{N}$. To enforce the validity conditions required by Theorem 4.2, we apply specific activation functions to these heads, as detailed below:

$$\boldsymbol{\mu}_{t_i} = \mathbf{f}_{\boldsymbol{\mu}}(\mathbf{h}_{t_i}), \quad \boldsymbol{\sigma}_{t_i} = \exp\left(\mathbf{f}_{\boldsymbol{\sigma}}(\mathbf{h}_{t_i})\right),$$
$$\boldsymbol{\gamma}_{t_i, t_{i+1}} = \exp\left(-\mathbf{f}_{\boldsymbol{\gamma}}(\mathbf{h}_{t_i}, \mathbf{h}_{t_{i+1}}) \cdot \Delta_{i,i+1}\right). \quad (15)$$

Once $\{\boldsymbol{\mu}_{t_i}, \boldsymbol{\sigma}_{t_i}, \boldsymbol{\gamma}_{t_i,t_{i+1}}\}_{i=0}^{N}$ is estimated, the underlying joint Gaussian structure of the variational posterior is fully determined. This allows us to learn the neural network parameters by maximizing $\mathcal{L}_{\text{marg}}(\boldsymbol{\theta}, \boldsymbol{\phi})$ where $\boldsymbol{\theta}, \boldsymbol{\phi}$ denotes the prior and approximate posterior parameters, respectively. To enable the differentiable sampling required for this optimization, we utilize the fact that the conditional transition $q(\mathbf{z}_{t_{i+1}}|\mathbf{z}_{t_i})$ is analytically derived as a Gaussian $\mathcal{N}(\mathbf{z}_{t_{i+1}}; \boldsymbol{\mu}_{t_{i+1}|t_i}, \boldsymbol{\Sigma}_{t_{i+1}|t_i})$, which is fully specified by the parameters $\{\boldsymbol{\mu}_{t_i}, \boldsymbol{\sigma}_{t_i}, \boldsymbol{\gamma}_{t_i,t_{i+1}}\}_{i=0}^{N}$ as seen from Eq. (10) and Eq. (11). Consequently, we can perform the standard reparameterization trick (Kingma & Welling, 2013) to sample the trajectory $\{\tilde{\mathbf{z}}_{t_i}\}_{i=0}^{N}$ recursively. Specifically, given a sample $\tilde{\mathbf{z}}_{t_i}$ at time $t_i$, the subsequent state $\tilde{\mathbf{z}}_{t_{i+1}}$ is generated via the following transition:

$$\tilde{\mathbf{z}}_{t_{i+1}} = \boldsymbol{\mu}_{t_{i+1}} + \boldsymbol{\gamma}_{t_i,t_{i+1}} \odot (\boldsymbol{\sigma}_{t_{i+1}} \oslash \boldsymbol{\sigma}_{t_i}) \odot (\tilde{\mathbf{z}}_{t_i} - \boldsymbol{\mu}_{t_i})$$
$$+ \boldsymbol{\sigma}_{t_{i+1}} \odot \sqrt{1 - \boldsymbol{\gamma}_{t_i,t_{i+1}}^2} \odot \boldsymbol{\epsilon}_{i+1}, \quad (16)$$

where $\boldsymbol{\epsilon}_{i+1} \sim \mathcal{N}(\mathbf{0}, \mathbf{I})$. With the sampling mechanism established, we can evaluate the tractable training objective:

$$\mathcal{L}_{\text{marg}}(\boldsymbol{\phi}) = \mathbb{E}_{q_{\boldsymbol{\phi}}}\left[\log p_{\boldsymbol{\theta}}(\mathcal{X}, \mathcal{Z})\right] + \mathcal{H}(q_{\boldsymbol{\phi}}(\mathcal{Z})). \quad (17)$$

The first term on the expectation of log-likelihood can be efficiently estimated via standard reparameterization sampling $\{\tilde{\mathbf{z}}_{t_i}\}_{i=0}^{N} \sim q_{\boldsymbol{\phi}}(\mathcal{Z})$ using the recursive transition in (16). Specifically, the joint density $p_{\boldsymbol{\theta}}(\mathcal{X}, \mathcal{Z})$ factorizes into the prior dynamics $p_{\boldsymbol{\theta}}(\mathcal{Z})$ and the emission likelihood $p_{\boldsymbol{\theta}}(\mathcal{X}|\mathcal{Z}) = \prod_{i=0}^{N} p_{\boldsymbol{\theta}}(\mathbf{x}_{t_i}|\mathbf{z}_{t_i})$. Each local likelihood is computed as a Gaussian $\mathcal{N}(\mathbf{x}_{t_i}; \mathbf{e}_{\boldsymbol{\theta}}(\mathbf{z}_{t_i}), \sigma^2\mathbf{I})$ and evaluated directly at each discrete timestamp $t_i$ using the sampled state $\tilde{\mathbf{z}}_{t_i}$. As for the prior term $\log p_{\boldsymbol{\theta}}(\mathcal{Z})$, leveraging the Markov property, the joint prior factorizes into $p_{\boldsymbol{\theta}}(\mathcal{Z}) = p_{\boldsymbol{\theta}}(\mathbf{z}_{t_0})\prod_{i=0}^{N-1} p_{\boldsymbol{\theta}}(\mathbf{z}_{t_{i+1}}|\mathbf{z}_{t_i})$, where $p_{\boldsymbol{\theta}}(\mathbf{z}_{t_0}) = \mathcal{N}(\boldsymbol{\mu}_{t_0}, \boldsymbol{\Sigma}_{t_0})$. The transition probability $p_{\boldsymbol{\theta}}(\mathbf{z}_{t_{i+1}}|\mathbf{z}_{t_i})$, which is determined by the SDE in Eq. (1), is governed

by the Fokker-Planck equation (FPE):

$$\frac{\partial p}{\partial t} = -\sum_{j=1}^{D} \frac{\partial}{\partial z_j}[\mathbf{f}_{\boldsymbol{\theta}}^{j}(t, \mathbf{z})p] + \frac{1}{2}\sum_{j,k=1}^{D} \frac{\partial^2}{\partial z_j \partial z_k}[(\mathbf{L}_{\boldsymbol{\theta}}\mathbf{L}_{\boldsymbol{\theta}}^{\top})_{jk}p]. \quad (18)$$

For a linear prior drift $\mathbf{f}_{\boldsymbol{\theta}}(t, \mathbf{z}_t) = \mathbf{F}_{\boldsymbol{\theta}}(t)\mathbf{z}_t + \mathbf{b}_{\boldsymbol{\theta}}(t)$ and state-independent diffusion $\mathbf{L}_{\boldsymbol{\theta}}(t)$, the transition distribution is strictly Gaussian: $p_{\boldsymbol{\theta}}(\mathbf{z}_{t_{i+1}}|\mathbf{z}_{t_i}) = \mathcal{N}(\mathbf{z}_{t_{i+1}}; \boldsymbol{\mu}_{t_{i+1}|t_i}, \boldsymbol{\Sigma}_{t_{i+1}|t_i})$, which can be analytically solved via the Lyapunov differential equations in Eq. (12) and computed *in parallel*. When the prior SDE involves a non-linear drift $\mathbf{f}_{\boldsymbol{\theta}}(t, \mathbf{z}_t)$ or state-dependent diffusion $\mathbf{L}_{\boldsymbol{\theta}}(t, \mathbf{z}_t)$, the FPE becomes analytically intractable. A typical approach(Ansari et al., 2023) is to approximate the transition density as a Gaussian $\mathcal{N}(\boldsymbol{\mu}_{t_{i+1}}, \boldsymbol{\Sigma}_{t_{i+1}})$. Specifically, initialized with $\boldsymbol{\mu}_{t_i} = \mathbf{z}_{t_i}$ and $\boldsymbol{\Sigma}_{t_i} = \mathbf{0}$, the parameters tracking the first two moments are obtained by numerically integrating the following coupled ODEs over $[t_i, t_{i+1}]$:

$$\frac{d\boldsymbol{\mu}_t}{dt} = \mathbf{f}_{\boldsymbol{\theta}}(t, \boldsymbol{\mu}_t), \quad (19)$$

$$\frac{d\boldsymbol{\Sigma}_t}{dt} = \mathbf{J}_{\mathbf{f}_{\boldsymbol{\theta}}}\boldsymbol{\Sigma}_t + \boldsymbol{\Sigma}_t\mathbf{J}_{\mathbf{f}_{\boldsymbol{\theta}}}^{\top} + \mathbf{L}_{\boldsymbol{\theta}}(t, \boldsymbol{\mu}_t)\mathbf{L}_{\boldsymbol{\theta}}(t, \boldsymbol{\mu}_t)^{\top}, \quad (20)$$

where $\mathbf{J}_{\mathbf{f}_{\boldsymbol{\theta}}} = \nabla_{\mathbf{z}}\mathbf{f}_{\boldsymbol{\theta}}(t, \mathbf{z})|_{\mathbf{z}=\boldsymbol{\mu}_t}$ denotes the local Jacobian of the drift. This system can be efficiently solved using general-purpose numerical ODE solvers, thereby obtaining the Gaussian approximation of the transition density $p_{\boldsymbol{\theta}}(\mathbf{z}_{t_{i+1}}|\mathbf{z}_{t_i})$. Unlike prior methods that heavily restrict the prior SDE to linear drifts and state-independent diffusion, our approach allows for flexible prior specifications including nonlinear dynamics and state-dependent noise.

The second entropy term $\mathcal{H}(q_{\boldsymbol{\phi}})$ can be decomposed into local entropy terms $\mathcal{H}(q_{\boldsymbol{\phi}}(\mathcal{Z})) = \sum_{i=0}^{N-1}\mathcal{H}(q_{\boldsymbol{\phi}}(\mathbf{z}_{t_i}, \mathbf{z}_{t_{i+1}})) - \sum_{i=1}^{N-1}\mathcal{H}(q_{\boldsymbol{\phi}}(\mathbf{z}_{t_i}))$. These entropy terms admit analytical closed-form solutions, allowing for fast and low-variance gradient estimation (see Appendix A.8 for full derivations).

By shifting the learning target from the SDE parameters $\{\mathbf{F}_{\boldsymbol{\phi}}, \mathbf{b}_{\boldsymbol{\phi}}, \mathbf{L}_{\boldsymbol{\phi}}\}$ to the valid distribution parameters $\{\boldsymbol{\mu}_{t_i}, \boldsymbol{\sigma}_{t_i}, \boldsymbol{\gamma}_{t_i,t_{i+1}}\}_{i=0}^{N}$, our framework bypasses the contractive constraint $\mathbf{F}_{\boldsymbol{\phi}} \prec \mathbf{0}$. This allows the model to capture flexible scaling dynamics beyond exponential decay between adjacent timestamps, preserving long-term dependencies. Further discussion is provided in Appendix E.

### 4.3. Nonlinear-SDE-Induced Variational Inference

So far, we have established a framework to learn $\{\boldsymbol{\mu}_{t_i}, \boldsymbol{\sigma}_{t_i}, \boldsymbol{\gamma}_{t_i,t_{i+1}}\}_{i=0}^{N}$, while ensuring that the corresponding Gauss-Markov distribution $q_{\boldsymbol{\phi}}(\mathcal{Z})$ is induced by a valid linear SDE. However, two critical limitations remain when modeling complex real-world data: 1) *simple dynamics*: Since the SDE parameters $\{\mathbf{F}_{\boldsymbol{\phi}}, \mathbf{b}_{\boldsymbol{\phi}}, \mathbf{L}_{\boldsymbol{\phi}}\}$ are assumed to be diagonal, the transition dynamics is constrained to scaling

or decay; 2) *unimodal density*: The Gaussian assumption restrict any $q_\phi(\mathbf{z}_{t_i}|\mathcal{X})$ to a unimodal density. This limitation renders the model ill-suited for capturing multimodal distributions, such as distinct behaviors (e.g., walking vs. sitting) where multiple valid states exist. To bridge this gap, we propose to employ a stack of Conditional Normalizing Flows (Winkler et al., 2019; Ardizzone et al., 2018) to transform the base distribution $q_\phi(\mathcal{Z})$ into a more expressive one, i.e., applying the flow to every sample $\tilde{\mathbf{z}}_{t_i}$ for $i = 1, 2, \cdots, N$, separately. Although the flows are applied to discrete samples, the underlying continuous-time dynamics of the refined process can be rigorously characterized as a non-linear SDE via Itô's lemma:

**Proposition 4.3.** *Let $\mathbf{z}_{[0,T]}^{(0)}$ be the base process governed by the linear SDE in* (6). *Consider a smooth, invertible transformation $T_\phi(\cdot;t)$ that transforms the base state $\mathbf{z}_t^{(0)}$ to the target state $\mathbf{z}_t$ via $\mathbf{z}_t = T_\phi(\mathbf{z}_t^{(0)};t)$. Then, the transformed process $\mathbf{z}_{[0,T]}^{(0)}$ is governed by the following non-linear Itô SDE for $t \in [0, T]$:*

$$d\mathbf{z}_t = \mathbf{g}_\phi(t, \mathbf{z}_t)\,dt + \mathbf{H}_\phi(t, \mathbf{z}_t)\,d\mathbf{W}_t, \qquad (21)$$

*with the drift $\mathbf{g}_\phi$ and diffusion $\mathbf{H}_\phi$ equal to:*

$$\mathbf{g}_\phi(t, \mathbf{z}) = \left[ \frac{\partial T_\phi}{\partial t} + \mathbf{J}_T \big( \mathbf{F}_\phi(t)\mathbf{z}^{(0)} + \mathbf{b}_\phi(t) \big) + \boldsymbol{\xi} \right]_{\mathbf{z}^{(0)}},$$
$$\mathbf{H}_\phi(t, \mathbf{z}) = \left[ \mathbf{J}_T \mathbf{L}_\phi(t) \right]_{\mathbf{z}^{(0)}},$$
$$(22)$$

*where $[\cdot]_{\mathbf{z}^{(0)}}$ denotes evaluation at the inverse map $\mathbf{z}^{(0)} = T_\phi^{-1}(\mathbf{z};t)$, and $\mathbf{J}_T = \nabla_{\mathbf{z}^{(0)}} T_\phi$ is the Jacobian matrix. The Itô correction term $\boldsymbol{\xi} \in \mathbb{R}^D$ accounts for the curvature of the transformation, with components defined by $\xi^k = \frac{1}{2} \mathrm{Tr} \left( \mathbf{L}_\phi(t)\mathbf{L}_\phi(t)^\top \nabla^2 T_\phi^k \right)$.*

In our implementation, we adopt the Glow architecture (Kingma & Dhariwal, 2018) by stacking $K$ invertible blocks. Each block incorporates ActNorm for stable training and Invertible $1 \times 1$ Convolution to couple latent dimensions. Crucially, we extend the standard affine coupling layer (Ardizzone et al., 2019) to be time-dependent. Instead of learning a static mapping, we inject the time-rich context $\mathbf{h}_t$ directly into the transformation to be time-dependent. Formally, let $\mathbf{z}_t^{(k-1)}$ denote the input to the $k$-th block at time $t$. We split it along the channel dimension into two partitions $\mathbf{z}_{a,t}^{(k-1)}$ and $\mathbf{z}_{b,t}^{(k-1)}$. The conditional affine transformation is then designed as:

$$\mathbf{z}'^{(k-1)}_{b,t} = \mathbf{z}_{b,t}^{(k-1)} \odot \exp\left( \mathbf{s}_\psi(\mathbf{z}_{a,t}^{(k-1)}, \mathbf{h}_t) \right) + \mathbf{t}_\psi(\mathbf{z}_{a,t}^{(k-1)}, \mathbf{h}_t),$$
$$(23)$$

where the scaling $\mathbf{s}_\psi$ and translation $\mathbf{t}_\psi$ are parameterized by neural networks conditioned on the context $\mathbf{h}_t$. The partitions $\mathbf{z}_{a,t}^{(k-1)}$ and $\mathbf{z}'^{(k-1)}_{b,t}$ are subsequently concatenated

to form the input for the next layer $\mathbf{z}_t^{(k)}$ (or the final output). Through the interplay of effective latent dimensions mixing via invertible $1 \times 1$ convolutions and the non-linear affine transformations, the unimodal base distribution is progressively mapped to a complex, time-evolving multimodal density. Further details are provided in Appendix C.2.

According to the change of variable theorem, flows introduce a log-determinant Jacobian term to the evaluation of $\mathcal{L}_{\mathrm{marg}}$ in (17). Specifically, the entropy term factorizes into $\mathcal{H}(q_\phi(\mathbf{z})) = \mathcal{H}(q_{\mathrm{base}}) + \mathbb{E}_{q_{\mathrm{base}}} \left[ \sum_{t \in \mathcal{T}} \sum_{k=1}^{K} \log \left| \det \frac{\partial \mathbf{z}_t^{(k)}}{\partial \mathbf{z}_t^{(k-1)}} \right| \right]$, where $\mathbf{z}_t^{(k)}$ denotes the intermediate state at flow layer $k$. For affine coupling layers, the log-determinant term further simplifies to the sum of log-scaling factors $\sum \log |s_{\psi_k}(\cdot)|$.

## 4.4. Complex-valued Inference for Oscillation Modeling

Although the flow-transformed distribution $q_\phi(\mathcal{Z})$ is highly expressive and guaranteed to be induced by a valid SDE, we recognize that the flows directly refine the density $q_\phi(\mathbf{z}_{t_i}|\mathcal{X})$ at each timestamp. Hence, the transition dynamics are refined only implicitly. However, explicitly decoupling dynamic patterns and incorporating inductive biases for oscillatory modes (e.g., periodic or quasi-periodic behaviors) has proven critical for modeling real-world time series (Wu et al., 2023; Lin et al., 2024; Deng et al., 2024; Zheng et al., 2025). According to spectral theory (Koopman, 1931; Takeishi et al., 2017; Lusch et al., 2018), the behavior of linear systems is governed by eigenvalues $\lambda = \alpha + i\beta$, where the real part $\alpha$ dictates scaling or decay, and the imaginary part $\beta$ governs oscillation. As a result, real-valued diagonal LTV-SDEs in (6) are inherently restricted to modeling scaling or decay. Therefore, we generalize our variational inference framework to the complex domain $\mathbb{C}^{D/2}$, explicitly capturing oscillation governed by imaginary components.

Specifically, we assume a complex diagonal structure akin to (6), where the drift $\mathbf{F}_\phi(t) = \mathrm{diag}(\boldsymbol{\alpha}(t) + i\boldsymbol{\beta}(t))$ are complex-valued, while diffusion $\mathbf{L}_\phi(t)$ remain real-valued. Generalizing Theorem 4.2 to the complex domain yields:

**Theorem 4.4** (Existence of a complex valid SDE). *The discrete Gauss-Markov distribution parameterized by $\{\boldsymbol{\mu}_{t_i}, \boldsymbol{\sigma}_{t_i}, \boldsymbol{\gamma}_{t_i,t_{i+1}}\}_{i=0}^{N}$ is induced by a valid complex-valued diagonal LTV-SDE if and only if the following conditions hold: 1) $\boldsymbol{\mu}(t) \in \mathbb{C}^{D/2}$ and $\boldsymbol{\sigma}(t) \in \mathbb{R}^{D/2}$ are continuously differentiable functions with $\boldsymbol{\sigma}(t) \succ \mathbf{0}$; 2) for any adjacent $t_i, t_{i+1}$ with interval $\Delta t_{i,i+1} \triangleq t_{i+1} - t_i$, the complex correlation coefficient $\boldsymbol{\gamma}_{t_i,t_{i+1}}$ can be written as:*

$$\boldsymbol{\gamma}_{i,i+1} = \exp\left( \bar{\mathbf{s}} \cdot \Delta t_{i,i+1} \right) \odot \exp\left( i\bar{\boldsymbol{\omega}} \cdot \Delta t_{i,i+1} \right), \quad (24)$$

*where $\bar{\mathbf{s}}(t_i, t_{i+1}) \succ \mathbf{0}$, $\bar{\boldsymbol{\omega}}(t_i, t_{i+1})$ are bounded continuous real functions.*

Compared to the real-valued case, this requires predicting

only one additional parameter: the average angular frequency $\bar{\omega}_i$ via a network head $\mathbf{f}_{\bar{\omega}}$. The sampling step in (16) generalizes to complex arithmetic with $\boldsymbol{\epsilon} \sim \mathcal{CN}(\mathbf{0}, \mathbf{I})$.

Although we parameterize the posterior via $D/2$ independent complex SDEs, each complex component $dz = (\alpha + i\beta)zdt + ldW_{\mathbb{C}}$ is mathematically isomorphic to a $2 \times 2$ real block-diagonal system:

$$d \begin{bmatrix} \mathrm{Re}(z) \\ \mathrm{Im}(z) \end{bmatrix} = \begin{bmatrix} \alpha & -\beta \\ \beta & \alpha \end{bmatrix} \begin{bmatrix} \mathrm{Re}(z) \\ \mathrm{Im}(z) \end{bmatrix} dt + \begin{bmatrix} l & 0 \\ 0 & l \end{bmatrix} d \begin{bmatrix} W_t^{\mathrm{Re}} \\ W_t^{\mathrm{Im}} \end{bmatrix}. \tag{25}$$

This structure implies that the posterior constitutes a block-diagonal Gaussian in the real space $\mathbb{R}^D$. In practice, we implement this using standard complex tensors for efficient sampling, concatenating the real and imaginary parts when feeding into downstream real-valued networks. Moreover, the nonlinear SDE-induced variational inference proposed in Sec. 4.3 can be directly extended to the complex domain. This yields a complex-valued nonlinear SDE-induced inference framework that incorporates oscillatory inductive biases while enabling flexible modeling of nonlinear and multimodal dynamics.

### 4.5. Efficient and Scalable Inference

**Parallel Inference.** By leveraging the parallelism of the Transformer encoder, the complete set of parameters $\{\boldsymbol{\mu}_{t_i}, \boldsymbol{\sigma}_{t_i}, \boldsymbol{\gamma}_{t_i, t_{i+1}}\}_{i=0}^N$ is generated simultaneously for all time steps. However, the sampling step derived in (16) constitutes a linear recurrence. To resolve this sequential bottleneck for ultra-long time series, we apply the parallel scan algorithm (Blelloch, 1990) in $O(\log N)$ time. See Appendix C.3 for details.

**Inherent Scalability.** Unlike most methods that require solving SDEs from the initial state $\mathbf{z}_0$, our formulation operates directly on local marginals. We assume that, given a local window $\mathcal{W}_k$ ($|\mathcal{W}_k| \ll |\mathcal{D}|$) with sufficient context, the Transformer encoder captures sufficient information for posterior inference, i.e., $q_\phi(\mathbf{z}|\mathcal{W}_k) \approx p(\mathbf{z}|\mathcal{D})$. This theoretically allows for a sliding window strategy where the global objective decomposes into a summation over local segments $\mathcal{L}(\theta, \phi; \mathcal{D}) \approx \sum_k \mathcal{L}(\theta, \phi; \mathcal{W}_k)$. This establishes a conceptual framework for future ultra-long sequence applications.

## 5. Experiments

**Tasks and Datasets** We evaluate SDE-VI against state-of-the-art baselines on four benchmarks: Human Activity, Pendulum, Physionet, and USHCN. These datasets cover diverse domains, ranging from IoT and physics simulations to healthcare and meteorology, each characterized by varying degrees of sparsity, irregularity, and distinct dynamic patterns. The evaluation covers four core tasks: point-wise

*Table 1.* Accuracy (%) for Human Activity Classification and Test MSE($\times 10^{-3}$) for Pendulum Regression. $^\dagger$Our implementation. Baseline results are taken from (Park et al., 2025).

| Model | Activity Acc ($\uparrow$) | Pendulum MSE ($\downarrow$) |
|---|---|---|
| Latent-ODE | $87.0 \pm 2.8$ | $15.70 \pm 0.29$ |
| CRU | $86.0 \pm 1.6^\dagger$ | $4.63 \pm 1.07$ |
| Latent-SDE$_\mathrm{H}$ | $90.6 \pm 0.4$ | $3.84 \pm 0.35$ |
| mTAND | $91.1 \pm 0.2$ | $3.20 \pm 0.60$ |
| ACSSM | $91.4 \pm 0.4$ | $2.98 \pm 0.30$ |
| **SDE-VI (Ours)** | $\mathbf{91.9 \pm 0.7}$ | $\mathbf{2.68 \pm 0.17}$ |

classification, sequence regression, sequence interpolation, and extrapolation. For a fair comparison, we adopt the pre-processing and splitting protocols from Park et al. (2025). Further details are provided in Appendix B.

**Baselines** (1) *RNNs:* GRU-$\Delta t$ and RKN-$\Delta t$, which are standard GRUs (Cho et al., 2014) and Recurrent Kalman Networks (RKNs)(Becker et al., 2019) augmented with the time gap $\Delta t$ to the next observation as input; and GRU-D (Che et al., 2016), which uses heuristic hidden state decay. (2) *Attention-based Model:* mTAND (Shukla & Marlin, 2021), a generative model employing multi-time cross-attention to transform between regular and irregular representations in the encoder and decoder. (3) *Neural ODEs:* ODE-RNN (Rubanova et al., 2019), Latent ODE (Chen et al., 2018), and GRU-ODE-B (De Brouwer et al., 2019). (4) *Neural SDEs:* CRU (Schirmer et al., 2022) and NCDSSM (Ansari et al., 2023) which employ Kalman filtering on linear prior SDEs; and two variational methods using approximate SDE: Latent-SDE$_\mathrm{H}$ (Zeng et al., 2023) with nonlinear SDEs, and ACSSM (Park et al., 2025) using locally LTI SDEs for both prior and posterior. A comprehensive theoretical and additional empirical comparison with other Neural SDEs is provided in Appendix D. We report the averaged results over three runs with different seeds. The best results are highlighted in **bold**, and the second-best results are shown in blue.

**Implementation Details** For implementation simplicity and a fair comparison with recent baselines (Park et al., 2025), we instantiate the prior SDE as a locally Linear Time-Invariant (LTI) system whose analytical transition probability $p_\theta(\mathbf{z}_{t_{i+1}}|\mathbf{z}_{t_i})$ allows for efficient calculation. Moreover, we follow the architectural setup of Park et al. (2025), employing the same modality-specific encoder for input embeddings, Transformer-based encoder for data assimilation and a shallow multi-layer perceptron (MLP) emission. See Appendix C for further details. Code is available at: https://github.com/SamoyedY/SDE-VI.

*Table 2.* Test MSE ($\times 10^{-2}$) for inter/extra-polation on Physionet and USHCN. Baselines are taken from (Park et al., 2025).

| Model | Interpolation | | Extrapolation | |
|---|---|---|---|---|
| | Physionet | USHCN | Physionet | USHCN |
| mTAND | $0.208 \pm 0.025$ | $1.766 \pm 0.009$ | $0.340 \pm 0.020$ | $2.360 \pm 0.038$ |
| RKN-$\Delta_t$ | $0.186 \pm 0.030$ | $0.009 \pm 0.002$ | $0.703 \pm 0.050$ | $1.491 \pm 0.272$ |
| GRU-$\Delta_t$ | $0.271 \pm 0.057$ | $0.090 \pm 0.059$ | $0.870 \pm 0.077$ | $2.081 \pm 0.054$ |
| GRU-D | $0.338 \pm 0.027$ | $0.944 \pm 0.011$ | $0.873 \pm 0.071$ | $1.718 \pm 0.015$ |
| Latent-ODE | $0.212 \pm 0.027$ | $1.798 \pm 0.009$ | $0.725 \pm 0.072$ | $2.034 \pm 0.005$ |
| ODE-RNN | $0.236 \pm 0.009$ | $0.831 \pm 0.008$ | $0.467 \pm 0.006$ | $1.955 \pm 0.466$ |
| GRU-ODE-B | $0.521 \pm 0.038$ | $0.841 \pm 0.142$ | $0.798 \pm 0.071$ | $5.437 \pm 1.020$ |
| CRU | $0.182 \pm 0.091$ | $0.016 \pm 0.006$ | $0.629 \pm 0.093$ | $1.273 \pm 0.066$ |
| ACSSM | $0.116 \pm 0.011$ | $\mathbf{0.006 \pm 0.001}$ | $0.627 \pm 0.019$ | $0.941 \pm 0.014$ |
| **SDE-VI (Ours)** | $\mathbf{0.052 \pm 0.002}$ | $0.044 \pm 0.012$ | $\mathbf{0.335 \pm 0.006}$ | $\mathbf{0.455 \pm 0.003}$ |

## 5.1. Point-wise Classification and Sequence Regression

**Human Activity Classification**   We evaluate classification performance using the Human Activity dataset, which comprises time-series data from five individuals performing various activities (e.g., walking, sitting). The dataset includes 12 features representing 3D positions captured by four sensors(belt, chest and both ankles). The objective is to classify the latent state of each time point into one of seven activities. Ideally, a latent state should encapsulate the complete information of the underlying system, yielding distinctive features for the classifier. Table 1 reports the test accuracy, showing SDE-VI outperforms all baseline models. While all methods except Latent-ODE and CRU use Transformers to capture contextual information, mTAND loses fine-grained continuous temporal structure during the conversion between regular and irregular points and path-based methods render the state non-Markovian. In contrast, SDE-VI captures fine-grained continuous-time dynamics while strictly maintaining the Markov property of the posterior, resulting in more accurate state inference.

**Pendulum Regression**   We then assess sequence regression performance on the pendulum experiment. The task involves inferring the sine and cosine of the pendulum angle from $24 \times 24$ pixel images observed at irregular intervals. The images are corrupted by a correlated noise process, challenging the model to capture temporally correlated stochasticity. The experimental results in Table 1 show that SDE-VI outperforms existing models. The poor performance of Latent-ODE highlights the necessity of modeling stochasticity with SDEs. Compared to other Neural SDEs, SDE-VI benefits from complex correlation coefficients that explicitly capture the oscillatory dynamics of the pendulum.

## 5.2. Sequence Interpolation and Extrapolation

**Datasets**   We evaluate the model on two healthcare and meteorological benchmarks. Physionet comprises 8,000 se-

quences of multivariate clinical records from ICU patients. Each sequence contains 37 time-variant features (e.g., vital signs), characterized by high sparsity (84% feature missing rate) and irregularity. USHCN contains daily meteorological measurements from 1,168 stations over four years. Although the original sampling is regular, the data is subsampled to simulate irregularity, yielding relatively stable but sparsely observed dynamics.

**Interpolation**   We mask a subset of observations and train the model to reconstruct the entire sequence. Table 2 shows that SDE-VI consistently outperforms all baseline models on Physionet in terms of MSE, achieving a significant performance gain over the second-best model. On USHCN, SDE-VI remains competitive with leading baselines (RKN, CRU, ACSSM). These models formulate latent evolution between adjacent observations as a convex combination of basis vectors, effectively imposing a structural prior that favors the smooth, stationary dynamics characteristic of USHCN. However, on the highly irregular Physionet dataset, such restrictive priors limit expressivity, whereas SDE-VI adapts flexibly to irregular variations.

**Extrapolation**   We further assess the model's ability to predict future observations based solely on history via causal attention. Compared to interpolation, extrapolation involves longer time intervals, requiring the model to maintain long-range dependencies. Table 2 shows that SDE-VI achieves state-of-the-art performance. Notably, mTAND also performs well with learnable multi-periodic embeddings that explicitly preserves long-range correlations. In contrast to linear SDE-based methods like CRU and ACSSM, which are constrained by stability-induced exponential decay assumptions, SDE-VI avoids such restrictions to adaptively capture long-range dependencies.

**Computational Efficiency**   We evaluate the computational efficiency of our method to demonstrate its significant

*Table 3.* Training runtime (s/epoch) and speedup relative to CRU.

| Model | USHCN | | Physionet | |
|---|---|---|---|---|
| | Time (↓) | Speedup (↑) | Time (↓) | Speedup (↑) |
| RKN-$\Delta_t$ | 32.1 | 1.3x | 114.9 | 2.6x |
| GRU-D | 99.6 | 0.4x | 2457.0 | 0.1x |
| Latent-ODE | 37.5 | 1.1x | 560.0 | 0.5x |
| ODE-RNN | 27.6 | 1.5x | 295.4 | 1.0x |
| GRU-ODE-B | 132.6 | 0.3x | 527.7 | 0.6x |
| CRU | 41.6 | 1.0x | 302.7 | 1.0x |
| ACSSM | 21.6 | 1.9x | 78.6 | 3.9x |
| **SDE-VI (Ours)** | 9.3 | 4.5x | 40.2 | 7.5x |
| *w/o Flow* | 9.6 | 4.3x | **37.2** | **8.1x** |
| *w/o Complex* | **9.0** | **4.6x** | 38.8 | 7.8x |

*Table 5.* Ablation study on different model components. EIP refers to the correlation parameterization in Theorem 4.2 and Theorem 4.4. Test MSE ($\times 10^{-2}$) for inter/extra-polation.

| Components | | | MSE (↓) | |
|---|---|---|---|---|
| EIP | Complex | Flow | Physionet (Inter.) | USHCN (Extra.) |
| ✗ | ✗ | ✗ | 0.094 | 0.709 |
| ✗ | ✓ | ✗ | 0.084 | 0.610 |
| ✗ | ✗ | ✓ | 0.075 | 0.537 |
| ✗ | ✓ | ✓ | 0.058 | 0.604 |
| ✓ | ✗ | ✗ | 0.084 | 0.572 |
| ✓ | ✓ | ✗ | 0.059 | 0.580 |
| ✓ | ✗ | ✓ | 0.065 | 0.485 |
| ✓ | ✓ | ✓ | **0.050** | **0.455** |

advantage. As shown in Table 3, SDE-VI achieves state-of-the-art efficiency, running substantially faster than CRU ($4.5\times \sim 7.5\times$) and outperforming ACSSM by approximately $2\times$. Furthermore, the ablation results (w/o Flow and w/o Complex) indicate that our enhancement modules introduce negligible computational overhead, confirming the high efficiency of our parallel framework.

## 5.3. Uncertainty Quantification

A fundamental advantage of SDE-based models is their capacity for principled uncertainty estimation. To evaluate whether the learned posteriors are well-calibrated, we expand our assessment beyond standard point-prediction metrics (e.g., MSE) to include uncertainty-focused evaluation. We report the predictive Negative Log-Likelihood (NLL) across all tasks. As summarized in Table 4, SDE-VI consistently achieves superior or highly competitive NLL scores compared to representative continuous-discrete state-space models. Experiments show that our framework not only provides accurate point forecasts but also infers highly expressive and well-calibrated posterior distributions.

*Table 4.* Test NLL (↓) across all tasks.

| Model | Activity | Pendulum | USHCN | | PhysioNet | |
|---|---|---|---|---|---|---|
| | | | Intra. | Extra. | Intra. | Extra. |
| CRU | 0.53 | -2.14 | **-5.34** | -4.15 | -4.25 | -3.96 |
| NCDSSM | 0.65 | -2.45 | -5.20 | -4.07 | -4.45 | -4.06 |
| ACSSM | 0.67 | -2.48 | -5.32 | -4.52 | -4.42 | -3.96 |
| **SDE-VI(Ours)** | **0.28** | **-2.55** | -5.31 | **-5.10** | **-4.60** | **-4.14** |

**Ablation Study** We conduct an ablation study on two representative tasks: interpolation on Physionet and extrapolation on USHCN. Table 5 summarizes the results, highlighting the individual efficiency of our proposed components.

Without EIP, correlation coefficients are directly parameterized by neural network as $\boldsymbol{\gamma}_{t_i, t_{i+1}} = \tanh(\mathbf{f}_{\boldsymbol{\gamma}}(\mathbf{h}_{t_i}, \mathbf{h}_{t_{i+1}}))$. This parameterization only ensures a valid Gauss-Markov distribution $q_\phi(\mathcal{Z})$, without guaranteeing it is induced by a

SDE. Results show EIP consistently improves performance, effectively translating the strict valid-SDE constraint on the variational distribution into substantial gains. Notably, this advantage is more pronounced on the extrapolation task, where SDE-VI with only EIP outperforms even the combination of Complex-valued Inference and flows (Rows 4 and 5). We then compare the individual contributions of flows and complex-valued inference. Overall, flows yield slightly larger gains, showing substantial improvements on both tasks. This stems from the ability of flows to refine linear dynamics and Gaussian distributions into more complex forms as shown in Theorem 4.3. Moreover, we investigate the hyperparameter sensitivity of the number of flow layers as detailed in Section C.6. While complex-valued inference introduces an inductive bias for oscillatory dynamics, offering limited benefits for stationary data like USHCN. Finally, using only a valid Gauss-Markov distribution (Row 1) already outperform other baselines on both tasks. This highlights the superiority of our marginal-based framework.

## 6. Conclusion

We proposed SDE-VI, a principled variational inference framework for generative modeling of irregular time series. By performing inference directly on posterior marginals induced by SDE, SDE-VI enables efficient and scalable inference beyond path-based models. The framework naturally extends to nonlinear, multimodal, and oscillatory dynamics through conditional normalizing flows and complex-valued generalization. Experiments across diverse domains show state-of-the-art performance in various downstream tasks.

## Acknowledgements

This work is supported by the National Natural Science Foundation of China (No. 62276280), Guangzhou Science and Technology Planning Project (No. 2024A04J9967). The authors would like to thank National Supercomputer Center in Guangzhou for providing high performance computational resources.

## Impact Statement

This paper advances the modeling of irregular time series. Our experimental results on medical datasets suggest that this framework could improve the reliability of automated diagnostic tools, thereby benefiting public health. We are not aware of direct negative societal impacts; however, practical application of these models requires careful ethical consideration regarding patient data privacy and algorithmic fairness.

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

# A. Theoretical Proofs and Derivations

In this section, we provide the detailed derivations and proofs for the theoretical claims made in the main text.

## A.1. Proof of Theorem 4.1 (The Hierarchy of ELBOs)

*Proof.* Recall the fundamental identity connecting the Evidence Lower Bound (ELBO) and the Kullback-Leibler divergence to the true posterior:

$$\mathcal{L}(\mathbb{Q}) = \log p_\theta(\mathcal{X}) - D_{\mathrm{KL}}(\mathbb{Q}||\mathbb{P}^*), \tag{26}$$

where $\mathbb{P}^* = \mathbb{P}_\theta(\cdot|\mathcal{X})$ denotes the true posterior path measure. Since the marginal log-likelihood $\log p_\theta(\mathcal{X})$ is constant with respect to the variational parameters, maximizing the ELBO is equivalent to minimizing the divergence from the true posterior.

We provide two proofs for the inequality $\mathcal{L}_{\mathrm{marg}}(q) \geq \mathcal{L}_{\mathrm{path}}(\mathbb{Q})$, with strict inequality holding whenever the true posterior dynamics deviate from the linear variational ansatz.

**Method 1: Information Theoretic Perspective (via Data Processing Inequality).** Let $\Omega = C([0, T], \mathbb{R}^D)$ be the continuous path space equipped with the Borel $\sigma$-algebra. Define the *projection mapping* $\pi_\mathcal{T} : \Omega \to \mathbb{R}^{(N+1) \times D}$ that restricts a path $\mathbf{z}_{0:T}$ to the observation timestamps: $\pi_\mathcal{T}(\mathbf{z}_{0:T}) = \mathcal{Z}$. The variational marginal $q$ and the true posterior marginal $p^*_{\mathrm{marg}}$ are precisely the *push-forward measures* of the path distributions $\mathbb{Q}$ and $\mathbb{P}^*$ under $\pi_\mathcal{T}$, respectively:

$$q = (\pi_\mathcal{T})_\# \mathbb{Q}, \quad p^*_{\mathrm{marg}} = (\pi_\mathcal{T})_\# \mathbb{P}^*. \tag{27}$$

The **Data Processing Inequality (DPI)** for $f$-divergences states that the KL divergence is non-increasing under any deterministic mapping. Applying DPI to $\pi_\mathcal{T}$ yields:

$$D_{\mathrm{KL}}(q||p^*_{\mathrm{marg}}) = D_{\mathrm{KL}}\big((\pi_\mathcal{T})_\# \mathbb{Q} \,||\, (\pi_\mathcal{T})_\# \mathbb{P}^*\big) \leq D_{\mathrm{KL}}(\mathbb{Q}||\mathbb{P}^*). \tag{28}$$

ality holds if and only if the sufficient statistic condition is met. However, $\mathbb{Q}$ is induced by a linear SDE (a Gaussian process), whereas $\mathbb{P}^*$ is induced by a non-linear SDE (typically a non-Gaussian process due to the neural likelihood). The projection $\pi_\mathcal{T}$ discards the fine-grained dynamics between observations (the *bridges*). Since the linear variational bridge cannot perfectly match the non-linear true posterior bridge almost everywhere, $\pi_\mathcal{T}$ is not sufficient for distinguishing $\mathbb{Q}$ from $\mathbb{P}^*$. Consequently, the inequality is strict:

$$D_{\mathrm{KL}}(q||p^*_{\mathrm{marg}}) < D_{\mathrm{KL}}(\mathbb{Q}||\mathbb{P}^*). \tag{29}$$

Substituting these terms back into the identity $\mathcal{L} = \log p(\mathcal{X}) - D_{\mathrm{KL}}$, we conclude $\mathcal{L}_{\mathrm{marg}}(q) > \mathcal{L}_{\mathrm{path}}(\mathbb{Q})$.

**Method 2: Explicit Decomposition (via Disintegration).** We derive the exact gap term $\Delta = \mathcal{L}_{\mathrm{marg}}(q) - \mathcal{L}_{\mathrm{path}}(\mathbb{Q})$. By the distribution rule for KL divergence (based on the Disintegration Theorem), the path divergence decomposes into a marginal term and the expected conditional bridge divergence:

$$D_{\mathrm{KL}}(\mathbb{Q}||\mathbb{P}^*) = D_{\mathrm{KL}}(q||p^*_{\mathrm{marg}}) + \mathbb{E}_{q(\mathcal{Z})}\left[D_{\mathrm{KL}}\big(\mathbb{Q}(\cdot|\mathcal{Z}) \,||\, \mathbb{P}^*(\cdot|\mathcal{Z})\big)\right], \tag{30}$$

where $\mathbb{Q}(\cdot|\mathcal{Z})$ denotes the pinned diffusion bridge measure conditioned on the discrete states $\mathcal{Z}$. Substituting this decomposition into the path-based ELBO formulation:

$$\begin{aligned}
\mathcal{L}_{\mathrm{path}}(\mathbb{Q}) &= \log p(\mathcal{X}) - \big(D_{\mathrm{KL}}(q||p^*_{\mathrm{marg}}) + \mathbb{E}_q\left[D_{\mathrm{KL}}(\mathbb{Q}(\cdot|\mathcal{Z})||\mathbb{P}^*(\cdot|\mathcal{Z}))\right]\big) \\
&= \underbrace{\big(\log p(\mathcal{X}) - D_{\mathrm{KL}}(q||p^*_{\mathrm{marg}})\big)}_{\mathcal{L}_{\mathrm{marg}}(q)} - \underbrace{\mathbb{E}_q\left[D_{\mathrm{KL}}(\mathbb{Q}(\cdot|\mathcal{Z})||\mathbb{P}^*(\cdot|\mathcal{Z}))\right]}_{\mathcal{E}_{\mathrm{bridge}} \geq 0}.
\end{aligned} \tag{31}$$

**Analysis of the Gap Term $\mathcal{E}_{\mathrm{bridge}}$.** This term quantifies the mismatch between the *linear variational bridge* and the *non-linear true posterior bridge*. Even if the marginals were perfectly optimized such that $q = p^*_{\mathrm{marg}}$, the linear ansatz of (6) constrains the conditional bridge $\mathbb{Q}(\cdot|\mathcal{Z})$ to be a Brownian bridge with linear drift. In contrast, the true posterior bridge $\mathbb{P}^*(\cdot|\mathcal{Z})$ is governed by the non-linear prior dynamics conditioned on observations. Thus, $\mathbb{Q}(\cdot|\mathcal{Z}) \neq \mathbb{P}^*(\cdot|\mathcal{Z})$ almost everywhere, implying $\mathcal{E}_{\mathrm{bridge}} > 0$.

Therefore, $\mathcal{L}_{\mathrm{marg}}(q)$ provides a strictly tighter lower bound on the log-likelihood than $\mathcal{L}_{\mathrm{path}}(\mathbb{Q})$ by avoiding the penalty term arising from the unavoidable bridge approximation error. □

## A.2. Derivation of the Correlation Coefficient $\gamma$ ( 13)

In this section, we derive the analytical relationship between the drift coefficient $\mathbf{F}_\phi(t)$ of the Linear Time-Varying (LTV) SDE and the discrete correlation coefficient $\gamma_{t_i,t_{i+1}}$ between two adjacent timestamps.

**1. Solution to the LTV-SDE.** Consider the variationally approximate SDE defined in (6):

$$d\mathbf{z}_t = [\mathbf{F}_\phi(t)\mathbf{z}_t + \mathbf{b}_\phi(t)]dt + \mathbf{L}_\phi(t)d\mathbf{W}_t. \tag{32}$$

Let $\boldsymbol{\Phi}(t,s)$ denote the *fundamental transition matrix* (or state transition matrix) of the homogeneous deterministic system $d\mathbf{z}_t = \mathbf{F}_\phi(t)\mathbf{z}_t dt$. Since $\mathbf{F}_\phi(t)$ is constrained to be diagonal for all $t$, the matrices $\mathbf{F}_\phi(t)$ and $\mathbf{F}_\phi(s)$ commute. Consequently, the transition matrix admits a closed-form exponential solution:

$$\boldsymbol{\Phi}(t,s) = \exp\left(\int_s^t \mathbf{F}_\phi(\tau)d\tau\right). \tag{33}$$

Note that $\boldsymbol{\Phi}(t,s)$ is also a diagonal matrix. Using the variation of constants formula, the solution to the SDE at time $t_{i+1}$ given the state at $t_i$ can be expressed as:

$$\mathbf{z}_{t_{i+1}} = \boldsymbol{\Phi}(t_{i+1}, t_i)\mathbf{z}_{t_i} + \int_{t_i}^{t_{i+1}} \boldsymbol{\Phi}(t_{i+1}, \tau)\mathbf{b}_\phi(\tau)d\tau + \int_{t_i}^{t_{i+1}} \boldsymbol{\Phi}(t_{i+1}, \tau)\mathbf{L}_\phi(\tau)d\mathbf{W}_\tau. \tag{34}$$

Let the noise term be denoted by $\boldsymbol{\epsilon}_{t_i,t_{i+1}} := \int_{t_i}^{t_{i+1}} \boldsymbol{\Phi}(t_{i+1}, \tau)\mathbf{L}_\phi(\tau)d\mathbf{W}_\tau$. Crucially, owing to the independent increments property of the Wiener process, $\boldsymbol{\epsilon}_{t_i,t_{i+1}}$ is independent of the current state $\mathbf{z}_{t_i}$.

**2. Cross-Covariance Propagation.** We define the centered state as $\tilde{\mathbf{z}}_t = \mathbf{z}_t - \mathbb{E}[\mathbf{z}_t]$. The cross-covariance matrix between $\mathbf{z}_{t_i}$ and $\mathbf{z}_{t_{i+1}}$ is defined as:

$$\mathbf{S}_{t_i,t_{i+1}} := \mathbb{E}\left[\tilde{\mathbf{z}}_{t_i}\tilde{\mathbf{z}}_{t_{i+1}}^\top\right]. \tag{35}$$

Substituting the SDE solution $\tilde{\mathbf{z}}_{t_{i+1}} = \boldsymbol{\Phi}(t_{i+1}, t_i)\tilde{\mathbf{z}}_{t_i} + \tilde{\boldsymbol{\epsilon}}_{t_i,t_{i+1}}$ into the definition:

$$\begin{aligned}
\mathbf{S}_{t_i,t_{i+1}} &= \mathbb{E}\left[\tilde{\mathbf{z}}_{t_i}\left(\boldsymbol{\Phi}(t_{i+1}, t_i)\tilde{\mathbf{z}}_{t_i} + \tilde{\boldsymbol{\epsilon}}_{t_i,t_{i+1}}\right)^\top\right] \\
&= \mathbb{E}\left[\tilde{\mathbf{z}}_{t_i}\tilde{\mathbf{z}}_{t_i}^\top \boldsymbol{\Phi}(t_{i+1}, t_i)^\top\right] + \underbrace{\mathbb{E}\left[\tilde{\mathbf{z}}_{t_i}\tilde{\boldsymbol{\epsilon}}_{t_i,t_{i+1}}^\top\right]}_{0} \\
&= \mathbf{S}_{t_i}\boldsymbol{\Phi}(t_{i+1}, t_i)^\top,
\end{aligned} \tag{36}$$

where $\mathbf{S}_{t_i} = \mathrm{Var}(\mathbf{z}_{t_i})$ is the marginal covariance at $t_i$. Since all involved matrices ($\mathbf{S}_{t_i}$, $\boldsymbol{\Phi}$) are diagonal, the transpose is identity, and matrix multiplication is commutative. Thus:

$$\mathbf{S}_{t_i,t_{i+1}} = \mathbf{S}_{t_i}\exp\left(\int_{t_i}^{t_{i+1}} \mathbf{F}_\phi(s)ds\right). \tag{37}$$

**3. Matching with Correlation Coefficient.** In our variational framework, the joint distribution is parameterized such that the cross-covariance relates to the correlation coefficient $\gamma_{t_i,t_{i+1}}$ via:

$$\mathbf{S}_{t_i,t_{i+1}} = \boldsymbol{\Sigma}_{t_i}\boldsymbol{\Gamma}_{t_i,t_{i+1}}\boldsymbol{\Sigma}_{t_{i+1}}, \tag{38}$$

where $\boldsymbol{\Sigma}_t = \mathbf{S}_t^{1/2} = \mathrm{diag}(\boldsymbol{s}_t)$ is the standard deviation matrix, and $\boldsymbol{\Gamma}_{t_i,t_{i+1}} = \mathrm{diag}(\boldsymbol{\gamma}_{t_i,t_{i+1}})$. Note that for diagonal Gaussians, $\mathbf{S}_{t_i,t_{i+1}}$ is symmetric. ating (37) and (38):

$$\mathbf{S}_{t_i}\exp\left(\int_{t_i}^{t_{i+1}} \mathbf{F}_\phi(s)ds\right) = \boldsymbol{\Sigma}_{t_i}\boldsymbol{\Gamma}_{t_i,t_{i+1}}\boldsymbol{\Sigma}_{t_{i+1}}. \tag{39}$$

Solving for the correlation matrix $\mathbf{\Gamma}_{t_i,t_{i+1}}$:

$$
\begin{aligned}
\mathbf{\Gamma}_{t_i,t_{i+1}} &= \mathbf{\Sigma}_{t_i}^{-1} \mathbf{S}_{t_i} \exp\left(\int_{t_i}^{t_{i+1}} \mathbf{F}_\phi(s)ds\right) \mathbf{\Sigma}_{t_{i+1}}^{-1} \\
&= \mathbf{\Sigma}_{t_i}^{-1} (\mathbf{\Sigma}_{t_i} \mathbf{\Sigma}_{t_i}) \exp\left(\int_{t_i}^{t_{i+1}} \mathbf{F}_\phi(s)ds\right) \mathbf{\Sigma}_{t_{i+1}}^{-1} \\
&= \mathbf{\Sigma}_{t_i} \exp\left(\int_{t_i}^{t_{i+1}} \mathbf{F}_\phi(s)ds\right) \mathbf{\Sigma}_{t_{i+1}}^{-1}.
\end{aligned}
\tag{40}
$$

Since all matrices are diagonal, this matrix equation holds element-wise for the vectors $\boldsymbol{\gamma}, \mathbf{s}, \mathbf{F}$. Rewriting in terms of Hadamard operations ($\odot$ for product, $\oslash$ for division):

$$
\boldsymbol{\gamma}_{t_i,t_{i+1}} = \boldsymbol{\sigma}_{t_i} \odot \exp\left(\int_{t_i}^{t_{i+1}} \mathbf{F}_\phi(s)ds\right) \oslash \boldsymbol{\sigma}_{t_{i+1}},
\tag{41}
$$

which concludes the proof.

### A.3. Proof of Theorem 4.2 (Existence of Valid SDE)

*Proof.* We establish the necessary and sufficient conditions for the existence of a valid diagonal LTV-SDE that induces the given discrete moments.

**1. Necessity ("Only if" Direction).** Assume the marginal distributions are induced by a valid diagonal SDE $d\mathbf{z}_t = (\mathbf{F}(t)\mathbf{z}_t + \mathbf{b}(t))dt + \mathbf{L}(t)d\mathbf{W}_t$, where the diffusion intensity $\mathbf{Q}(t) = \mathbf{L}(t)\mathbf{L}(t)^\top$ is non-negative and bounded. The correlation coefficient $\boldsymbol{\gamma}_{t_i,t_{i+1}}$ is analytically given by:

$$
\boldsymbol{\gamma}_{t_i,t_{i+1}} = \exp\left(\int_{t_i}^{t_{i+1}} \mathbf{F}(\tau)\,d\tau\right) \odot \sqrt{\mathbf{\Sigma}(t_i) \oslash \mathbf{\Sigma}(t_{i+1})}.
\tag{42}
$$

The variance $\mathbf{\Sigma}(t)$ evolves according to the Lyapunov differential equation $\dot{\mathbf{\Sigma}}(t) = 2\mathbf{F}(t) \odot \mathbf{\Sigma}(t) + \mathbf{Q}(t)$. Rearranging for the drift term (element-wise), we have:

$$
\mathbf{F}(t) = (\dot{\mathbf{\Sigma}}(t) - \mathbf{Q}(t)) \oslash 2\mathbf{\Sigma}(t).
\tag{43}
$$

Substituting this expression into the logarithm of the correlation formula:

$$
\begin{aligned}
\ln \boldsymbol{\gamma}_{t_i,t_{i+1}} &= \int_{t_i}^{t_{i+1}} \mathbf{F}(\tau)\,d\tau + \frac{1}{2}\left(\ln \mathbf{\Sigma}(t_i) - \ln \mathbf{\Sigma}(t_{i+1})\right) \\
&= \int_{t_i}^{t_{i+1}} \left(\frac{\dot{\mathbf{\Sigma}}(\tau)}{2\mathbf{\Sigma}(\tau)} - \frac{\mathbf{Q}(\tau)}{2\mathbf{\Sigma}(\tau)}\right) d\tau - \frac{1}{2}\Delta \ln \mathbf{\Sigma} \\
&= \frac{1}{2}\int_{t_i}^{t_{i+1}} d(\ln \mathbf{\Sigma}(\tau)) - \int_{t_i}^{t_{i+1}} \frac{\mathbf{Q}(\tau)}{2\mathbf{\Sigma}(\tau)}\,d\tau - \frac{1}{2}\Delta \ln \mathbf{\Sigma} \\
&= -\int_{t_i}^{t_{i+1}} \frac{\mathbf{Q}(\tau)}{2\mathbf{\Sigma}(\tau)}\,d\tau.
\end{aligned}
\tag{44}
$$

The boundary variance terms cancel out perfectly. By the Mean Value Theorem for Integrals (applied element-wise), there exists an average rate vector $\bar{\mathbf{s}}_i$ such that the integral equals $\bar{\mathbf{s}}_i \Delta t_{i,i+1}$. Since $\mathbf{Q}(\tau) \succeq \mathbf{0}$ and $\mathbf{\Sigma}(\tau) \succ \mathbf{0}$, the integrand is non-negative, implying $\bar{\mathbf{s}}_i \succeq \mathbf{0}$. For a non-degenerate process ($\mathbf{Q} \succ \mathbf{0}$), we strictly have $\bar{\mathbf{s}}_i \succ \mathbf{0}$. Thus, $\boldsymbol{\gamma}_{t_i,t_{i+1}} = \exp(-\bar{\mathbf{s}}_i \Delta t)$.

**2. Sufficiency ("If" Direction via Construction).** Given continuous moments $\{\boldsymbol{\mu}(t), \mathbf{\Sigma}(t)\}$ and correlations satisfying $\boldsymbol{\gamma}_{t_i,t_{i+1}} = \exp(-\bar{\mathbf{s}}_i \Delta t)$ with bounded $\bar{\mathbf{s}}_i \succ \mathbf{0}$, we construct a valid underlying SDE via a piecewise Linear Time-Invariant (LTI) approximation on $[t_i, t_{i+1}]$. We define the integral operator for a diagonal matrix $\mathbf{A}$ as $\mathcal{E}(\mathbf{A}) \triangleq \int_0^{\Delta t} e^{\mathbf{A}(\Delta t - \tau)}\,d\tau = (e^{\mathbf{A}\Delta t} - \mathbf{I}) \oslash \mathbf{A}$.

**Step 1: Constructing the Drift Slope $\bar{\mathbf{F}}_i$.** We define the effective drift slope $\bar{\mathbf{F}}_i$ to satisfy the correlation transport equation:

$$e^{\bar{\mathbf{F}}_i \Delta t} \triangleq \boldsymbol{\gamma}_{t_i, t_{i+1}} \odot \sqrt{\boldsymbol{\Sigma}(t_{i+1}) \oslash \boldsymbol{\Sigma}(t_i)}. \tag{45}$$

**Step 2: Constructing the Drift Bias $\bar{\mathbf{b}}_i$.** The mean of an LTI system evolves as $\boldsymbol{\mu}(t_{i+1}) = e^{\bar{\mathbf{F}}_i \Delta t} \odot \boldsymbol{\mu}(t_i) + \mathcal{E}(\bar{\mathbf{F}}_i) \odot \bar{\mathbf{b}}_i$. Solving for the constant bias $\bar{\mathbf{b}}_i$ to match the target mean:

$$\bar{\mathbf{b}}_i = \left( \boldsymbol{\mu}(t_{i+1}) - e^{\bar{\mathbf{F}}_i \Delta t} \odot \boldsymbol{\mu}(t_i) \right) \oslash \mathcal{E}(\bar{\mathbf{F}}_i). \tag{46}$$

Note that $\mathcal{E}(\bar{\mathbf{F}}_i)$ is strictly positive (as $e^{\mathbf{A}\tau} > 0$), ensuring $\bar{\mathbf{b}}_i$ is well-defined.

**Step 3: Constructing the Diffusion $\mathbf{Q}_i$.** The variance of an LTI system evolves as $\boldsymbol{\Sigma}(t_{i+1}) = e^{2\bar{\mathbf{F}}_i \Delta t} \odot \boldsymbol{\Sigma}(t_i) + \mathbf{Q}_i \odot \mathcal{E}(2\bar{\mathbf{F}}_i)$. Note the coefficient is $2\bar{\mathbf{F}}_i$. Substituting the definition of $\bar{\mathbf{F}}_i$ from Step 1, we have $e^{2\bar{\mathbf{F}}_i \Delta t} \odot \boldsymbol{\Sigma}(t_i) = \boldsymbol{\gamma}_{t_i, t_{i+1}}^2 \odot \boldsymbol{\Sigma}(t_{i+1}) = e^{-2\bar{\mathbf{s}}_i \Delta t} \odot \boldsymbol{\Sigma}(t_{i+1})$. Solving for $\mathbf{Q}_i$:

$$\mathbf{Q}_i = \left( \boldsymbol{\Sigma}(t_{i+1}) \odot (\mathbf{1} - e^{-2\bar{\mathbf{s}}_i \Delta t}) \right) \oslash \mathcal{E}(2\bar{\mathbf{F}}_i). \tag{47}$$

Since $\bar{\mathbf{s}}_i \succ \mathbf{0}$ implies $(\mathbf{1} - e^{-2\bar{\mathbf{s}}_i \Delta t}) \succ \mathbf{0}$, and $\mathcal{E}(\cdot) \succ \mathbf{0}$, we guarantee a valid diffusion intensity $\mathbf{Q}_i \succ \mathbf{0}$.

**Limit Analysis ($\Delta t \to 0$).** We verify that the constructed parameters correspond to a valid SDE with finite coefficients in the continuous limit. We employ first-order Taylor expansions with Peano remainder terms: $e^{\mathbf{A}\Delta t} = \mathbf{I} + \mathbf{A}\Delta t + O(\Delta t^2)$ and $\mathcal{E}(\mathbf{A}) = \Delta t \cdot \mathbf{I} + O(\Delta t^2)$.

**1. Drift Bias Consistency:**

$$\lim_{\Delta t \to 0} \bar{\mathbf{b}}_i = \lim_{\Delta t \to 0} \frac{\boldsymbol{\mu}(t + \Delta t) - (\mathbf{I} + \bar{\mathbf{F}}_i \Delta t)\boldsymbol{\mu}(t)}{\Delta t} \tag{48}$$
$$= \dot{\boldsymbol{\mu}}(t) - \bar{\mathbf{F}}(t) \odot \boldsymbol{\mu}(t).$$

This recovers the standard ODE $\dot{\boldsymbol{\mu}} = \mathbf{F}\boldsymbol{\mu} + \mathbf{b}$. Since $\boldsymbol{\mu}(t)$ is differentiable (by assumption of valid moments), $\mathbf{b}(t)$ is finite.

**2. Diffusion Intensity Consistency:**

$$\lim_{\Delta t \to 0} \mathbf{Q}_i = \lim_{\Delta t \to 0} \frac{\boldsymbol{\Sigma}(t + \Delta t) \odot (2\bar{\mathbf{s}}_i \Delta t + O(\Delta t^2))}{\Delta t + O(\Delta t^2)} \tag{49}$$
$$= 2\boldsymbol{\Sigma}(t) \odot \bar{\mathbf{s}}_i.$$

Since $\bar{\mathbf{s}}_i$ is bounded and $\boldsymbol{\Sigma}(t)$ is continuous, the derived diffusion intensity $\mathbf{Q}(t) = 2\boldsymbol{\Sigma}(t) \odot \bar{\mathbf{s}}(t)$ is finite and non-negative. Thus, the constructed triplet $\{\mathbf{F}(t), \mathbf{b}(t), \mathbf{Q}(t)\}$ constitutes a valid SDE system. $\qquad\square$

## A.4. Physical Interpretation: Entropy Increment Parameterization

We provide the physical justification for the parameterization $\boldsymbol{\gamma}_{t_i, t_{i+1}} = \exp(-\bar{\mathbf{s}}_i \Delta t)$ by proving that $\bar{\mathbf{s}}$ corresponds to the rate of entropy production in a normalized phase space.

*Proof.* The proof proceeds in two steps: first, we introduce a coordinate transformation to decouple the drift dynamics; second, we analyze the entropy evolution of the transformed state.

**1. The Martingale Transform (Decoupling Drift).** Consider the centered latent state $\tilde{\mathbf{z}}_t = \mathbf{z}_t - \boldsymbol{\mu}(t)$. Its dynamics are given by:

$$d\tilde{\mathbf{z}}_t = \mathbf{F}(t) \odot \tilde{\mathbf{z}}_t \, dt + \mathbf{L}(t) \, d\mathbf{W}_t. \tag{50}$$

We introduce a *normalized state* $\mathbf{y}_t$ that removes the deterministic drift influence via an integrating factor $\mathbf{E}(t) = \exp\left( - \int_0^t \mathbf{F}(s) ds \right)$:

$$\mathbf{y}_t \triangleq \mathbf{E}(t) \odot \tilde{\mathbf{z}}_t. \tag{51}$$

By Itô's Lemma, the drift term cancels strictly, and $\mathbf{y}_t$ evolves as a pure martingale:

$$d\mathbf{y}_t = (\mathbf{E}(t) \odot \mathbf{L}(t)) \; d\mathbf{W}_t. \tag{52}$$

The variance of this transformed state, denoted $\boldsymbol{\nu}(t) \triangleq \mathrm{Var}(\mathbf{y}_t)$, relates to the original variance $\boldsymbol{\Sigma}(t)$ via:

$$\boldsymbol{\nu}(t) = \mathbf{E}(t)^2 \odot \boldsymbol{\Sigma}(t) = e^{-2 \int_0^t \mathbf{F}(s)ds} \odot \boldsymbol{\Sigma}(t). \tag{53}$$

**2. Entropy Evolution and Correlation.** For a Gaussian variable, the differential entropy is proportional to the log-variance. We define the *Phase Space Entropy* $\mathcal{H}(t)$ of the normalized state $\mathbf{y}_t$ as:

$$\mathcal{H}(t) \triangleq \frac{1}{2} \ln \boldsymbol{\nu}(t) = \frac{1}{2} \ln \boldsymbol{\Sigma}(t) - \int_0^t \mathbf{F}(s) \, ds. \tag{54}$$

Note that this quantity $\mathcal{H}(t)$ measures the intrinsic uncertainty accumulated purely from diffusion, stripping away the expansion/contraction effects of the drift field $\mathbf{F}$. Differentiating $\mathcal{H}(t)$ with respect to time using the Lyapunov equation $\dot{\boldsymbol{\Sigma}} = 2\mathbf{F}\boldsymbol{\Sigma} + \mathbf{Q}$:

$$\dot{\mathcal{H}}(t) = \frac{1}{2} \frac{\dot{\boldsymbol{\Sigma}}}{\boldsymbol{\Sigma}} - \mathbf{F} = \frac{1}{2} \frac{2\mathbf{F}\boldsymbol{\Sigma} + \mathbf{Q}}{\boldsymbol{\Sigma}} - \mathbf{F} = \frac{1}{2} \frac{\mathbf{Q}(t)}{\boldsymbol{\Sigma}(t)}. \tag{55}$$

Crucially, the drift $\mathbf{F}$ cancels out. The rate of entropy production $\dot{\mathcal{H}}$ is strictly non-negative (since $\mathbf{Q}, \boldsymbol{\Sigma} \succ 0$), reflecting the irreversibility of the diffusion process.

Finally, we link this to the correlation coefficient. For the Linear SDE, the correlation is analytically given by (see 13):

$$\boldsymbol{\gamma}_{t_i, t_{i+1}} = \exp\left( \int_{t_i}^{t_{i+1}} \mathbf{F}(s)ds \right) \odot \sqrt{\frac{\boldsymbol{\Sigma}(t_i)}{\boldsymbol{\Sigma}(t_{i+1})}}. \tag{56}$$

Substituting the relation $\int \mathbf{F} ds = \frac{1}{2} \ln \boldsymbol{\Sigma} - \mathcal{H}$ from the entropy definition:

$$\begin{aligned}
\ln \boldsymbol{\gamma}_{t_i, t_{i+1}} &= \left( \frac{1}{2} \ln \boldsymbol{\Sigma}_{i+1} - \mathcal{H}_{i+1} - \frac{1}{2} \ln \boldsymbol{\Sigma}_i + \mathcal{H}_i \right) + \frac{1}{2} (\ln \boldsymbol{\Sigma}_i - \ln \boldsymbol{\Sigma}_{i+1}) \\
&= -(\mathcal{H}(t_{i+1}) - \mathcal{H}(t_i)) \\
&= - \int_{t_i}^{t_{i+1}} \dot{\mathcal{H}}(\tau) \, d\tau.
\end{aligned} \tag{57}$$

Thus, $\boldsymbol{\gamma}_{t_i, t_{i+1}} = \exp\left(-\Delta \mathcal{H}_{i, i+1}\right)$. Defining the average entropy rate $\bar{\mathbf{s}}_i \triangleq \frac{1}{\Delta t} \int_{t_i}^{t_{i+1}} \dot{\mathcal{H}}(\tau) d\tau$, we recover the parameterization:

$$\boldsymbol{\gamma}_{t_i, t_{i+1}} = \exp\left(-\bar{\mathbf{s}}_i \cdot \Delta t_{i, i+1}\right). \tag{58}$$

Since $\dot{\mathcal{H}} = \frac{\mathbf{Q}}{2\boldsymbol{\Sigma}} \succ 0$, it follows that $\bar{\mathbf{s}}_i \succ 0$. □

### A.5. Proof of Theorem 4.3: Dynamics of Flow-Induced SDE

*Proof.* We rigorously derive the stochastic differential equation governing the transformed latent variable $\mathbf{z}_t$, demonstrating that the flow-based refinement induces a valid non-linear diffusion process with state-dependent noise.

**Setup and Regularity Conditions.** Let the base process $\mathbf{z}_t^{(0)}$ be governed by the diagonal linear SDE defined in (6):

$$d\mathbf{z}_t^{(0)} = \boldsymbol{\mu}_{\text{base}}(t, \mathbf{z}_t^{(0)}) \, dt + \mathbf{L}_{\boldsymbol{\phi}}(t) \, d\mathbf{W}_t, \tag{59}$$

where the base drift is affine $\boldsymbol{\mu}_{\text{base}}(t, \mathbf{z}^{(0)}) = \mathbf{F}_{\boldsymbol{\phi}}(t)\mathbf{z}^{(0)} + \mathbf{b}_{\boldsymbol{\phi}}(t)$, and $\mathbf{Q}_{\boldsymbol{\phi}}(t) = \mathbf{L}_{\boldsymbol{\phi}}(t)\mathbf{L}_{\boldsymbol{\phi}}(t)^\top$ denotes the diffusion tensor.

Consider the refined variable $\mathbf{z}_t = T_{\boldsymbol{\phi}}(\mathbf{z}_t^{(0)}; t)$, where the transformation $T_{\boldsymbol{\phi}} : \mathbb{R}^D \times \mathcal{T} \to \mathbb{R}^D$ is assumed to be a $C^{2,1}$ diffeomorphism (twice continuously differentiable in state, once in time).

**Application of Multidimensional Itô's Lemma.** Applying the multidimensional Itô's formula to the vector-valued function $T_\phi(\mathbf{z}_t^{(0)}; t)$, the differential $d\mathbf{z}_t$ is given by:

$$d\mathbf{z}_t = \frac{\partial T_\phi}{\partial t} dt + \mathbf{J}_T(\mathbf{z}_t^{(0)}; t) d\mathbf{z}_t^{(0)} + \frac{1}{2}\boldsymbol{\xi}(\mathbf{z}_t^{(0)}; t) dt, \tag{60}$$

where $\mathbf{J}_T(\mathbf{z}^{(0)}; t) = \nabla_{\mathbf{z}^{(0)}} T_\phi \in \mathbb{R}^{D \times D}$ is the Jacobian matrix. The second-order correction term $\boldsymbol{\xi} \in \mathbb{R}^D$ arises from the quadratic covariations $d(\mathbf{z}_t^{(0)})^i d(\mathbf{z}_t^{(0)})^j = Q_\phi^{ij}(t) dt$. Its $k$-th component is defined by the trace of the diffusion-Hessian product:

$$\xi^k(\mathbf{z}^{(0)}; t) = \mathrm{Tr}\left(\mathbf{Q}_\phi(t)\nabla^2 T^k(\mathbf{z}^{(0)}; t)\right), \quad k \in \{1, \ldots, D\}, \tag{61}$$

where $\nabla^2 T^k$ is the Hessian matrix of the $k$-th output component of the transformation.

**Substitution and Induced Dynamics.** Substituting the dynamics of $d\mathbf{z}_t^{(0)}$ into the expression for $d\mathbf{z}_t$:

$$\begin{aligned}
d\mathbf{z}_t &= \frac{\partial T_\phi}{\partial t} dt + \mathbf{J}_T(\mathbf{z}_t^{(0)}; t)\left[\boldsymbol{\mu}_{\text{base}}(t, \mathbf{z}_t^{(0)}) dt + \mathbf{L}_\phi(t) d\mathbf{W}_t\right] + \frac{1}{2}\boldsymbol{\xi}(\mathbf{z}_t^{(0)}; t) dt \\
&= \underbrace{\left(\frac{\partial T_\phi}{\partial t} + \mathbf{J}_T(\mathbf{z}_t^{(0)}; t)\boldsymbol{\mu}_{\text{base}}(t, \mathbf{z}_t^{(0)}) + \frac{1}{2}\boldsymbol{\xi}(\mathbf{z}_t^{(0)}; t)\right)}_{\text{Induced Drift}} dt + \underbrace{\mathbf{J}_T(\mathbf{z}_t^{(0)}; t)\mathbf{L}_\phi(t)}_{\text{Induced Diffusion}} d\mathbf{W}_t.
\end{aligned} \tag{62}$$

To express the dynamics solely in terms of the target variable $\mathbf{z}_t$, we utilize the inverse mapping $\mathbf{z}_t^{(0)} = T_\phi^{-1}(\mathbf{z}_t; t)$. The induced SDE is thus:

$$d\mathbf{z}_t = \mathbf{g}_\phi(t, \mathbf{z}_t) dt + \mathbf{H}_\phi(t, \mathbf{z}_t) d\mathbf{W}_t, \tag{63}$$

where the coefficients are explicitly defined as:

$$\mathbf{H}_\phi(t, \mathbf{z}) = \left[\mathbf{J}_T(\mathbf{z}^{(0)}; t)\, \mathbf{L}_\phi(t)\right]_{\mathbf{z}^{(0)}=T_\phi^{-1}(\mathbf{z};t)}, \tag{64}$$

$$\mathbf{g}_\phi(t, \mathbf{z}) = \left[\frac{\partial T_\phi}{\partial t} + \mathbf{J}_T(\mathbf{z}^{(0)}; t)\left(\mathbf{F}_\phi(t)\mathbf{z}^{(0)} + \mathbf{b}_\phi(t)\right) + \frac{1}{2}\boldsymbol{\xi}(\mathbf{z}^{(0)}; t)\right]_{\mathbf{z}^{(0)}=T_\phi^{-1}(\mathbf{z};t)}. \tag{65}$$

This derivation confirms that the linear base dynamics are mathematically transformed into a non-linear drift $\mathbf{g}_\phi$ and a state-dependent diffusion $\mathbf{H}_\phi$, enabling the model to capture complex, heteroscedastic data distributions. $\quad\square$

### A.6. Proof of Theorem 4.4

*Proof.* We extend the entropy parameterization to the complex domain $\mathbb{C}^D$. We demonstrate that the complex correlation coefficient exhibits a canonical decoupling into a magnitude term (governed by entropy) and a phase term (governed by oscillatory drift).

**1. Complex SDE Formulation.** Let $z_t \in \mathbb{C}$ denote a generic component of the latent variable (omitting component indices for brevity). The dynamics follow a linear complex SDE:

$$dz_t = \lambda(t)z_t dt + \sigma(t) dW_t^{\mathbb{C}}, \tag{66}$$

where:

- The complex drift rate $\lambda(t) = \alpha(t) + i\beta(t)$ comprises a real expansion rate $\alpha(t)$ and an imaginary rotation rate (angular frequency) $\beta(t)$.

- The noise process is a standard complex Wiener process $W_t^{\mathbb{C}} = W_t^{\text{Re}} + iW_t^{\text{Im}}$, satisfying the quadratic variation rules $dW_t^{\text{Re}}dW_t^{\text{Re}} = dW_t^{\text{Im}}dW_t^{\text{Im}} = dt$.

- The diffusion intensity is isotropic. The total infinitesimal variance is given by the Hermitian quadratic variation $dz_t d\bar{z}_t = \sigma^2(t)(dW_t^{\mathbb{C}}d\overline{W}_t^{\mathbb{C}}) = 2\sigma^2(t) dt \triangleq Q(t) dt$.

**2. Evolution of Mean and Variance.** The mean $\mu_t = \mathbb{E}[z_t]$ evolves according to the ODE $\dot{\mu}_t = \lambda(t)\mu_t$. The transition factor over $[t_i, t_{i+1}]$ is simply $\mathbf{A}_i = \exp(\int_{t_i}^{t_{i+1}} \lambda(s)ds)$.

To derive the variance dynamics, let $\Sigma(t) = \mathbb{E}[|z_t - \mu_t|^2] = \text{Cov}(z_t, z_t)$. Applying the **Complex Itô Product Rule** to the centered squared modulus $|z - \mu|^2$:

$$
\begin{aligned}
d\Sigma(t) &= \mathbb{E}\left[ (z - \mu)d(\overline{z - \mu}) + (\overline{z - \mu})d(z - \mu) + d(z - \mu)d(\overline{z - \mu}) \right] \\
&= \left[ \Sigma(t)\overline{\lambda(t)} + \Sigma(t)\lambda(t) + Q(t) \right] dt \\
&= \left[ \Sigma(t)(\alpha - i\beta) + \Sigma(t)(\alpha + i\beta) + Q(t) \right] dt \\
&= \left( 2\alpha(t)\Sigma(t) + Q(t) \right) dt.
\end{aligned}
\tag{67}
$$

*Crucially, the imaginary rotational terms $\pm i\beta(t)$ cancel out perfectly.* This confirms that the variance envelope depends solely on the real expansion rate $\alpha(t)$, mirroring the real-valued case.

**3. Entropy and Correlation Decoupling.** Since the variance evolution is independent of $\beta(t)$, the *Reduced Phase Space Entropy* $\mathbf{S}(t)$ remains defined by the real drift: $\mathbf{S}(t) = \ln \Sigma(t) - 2 \int_0^t \alpha(s)ds$.

The complex correlation coefficient $\boldsymbol{\gamma}_{i,i+1}$ is defined as the normalized Hermitian cross-covariance:

$$
\boldsymbol{\gamma}_{i,i+1} = \frac{\text{Cov}(z_{t_{i+1}}, z_{t_i})}{\sqrt{\Sigma(t_{i+1})\Sigma(t_i)}}.
\tag{68}
$$

Using the solution to the linear SDE, the covariance is $\text{Cov}(z_{t_{i+1}}, z_{t_i}) = \mathbf{A}_i \Sigma(t_i)$. Substituting the polar form of the transition operator $\mathbf{A}_i = e^{\int \alpha} \cdot e^{i \int \beta}$:

$$
\begin{aligned}
\boldsymbol{\gamma}_{i,i+1} &= \left( e^{\int_{t_i}^{t_{i+1}} \alpha(s)ds} \cdot e^{i \int_{t_i}^{t_{i+1}} \beta(s)ds} \right) \cdot \frac{\Sigma(t_i)}{\sqrt{\Sigma(t_{i+1})\Sigma(t_i)}} \\
&= e^{i \int_{t_i}^{t_{i+1}} \beta(s)ds} \cdot \left( e^{\int_{t_i}^{t_{i+1}} \alpha(s)ds} \sqrt{\frac{\Sigma(t_i)}{\Sigma(t_{i+1})}} \right).
\end{aligned}
\tag{69}
$$

By the definition of entropy $\mathbf{S}(t)$, the real-valued term simplifies exactly as in Theorem 4.2, eliminating the drift integral $\int \alpha$. We are left with the final decoupled form:

$$
\boldsymbol{\gamma}_{i,i+1} = \underbrace{\exp\left( \frac{\mathbf{S}(t_i) - \mathbf{S}(t_{i+1})}{2} \right)}_{\text{Amplitude Decay}} \odot \underbrace{\exp\left( i \int_{t_i}^{t_{i+1}} \beta(s)\, ds \right)}_{\text{Phase Rotation}}.
\tag{70}
$$

Thus, the magnitude of the correlation is controlled purely by entropy change, while the phase is controlled purely by the accumulated rotation. $\qquad\square$

### A.7. Derivation of Complex LTI Coefficients

In this section, we derive the explicit coefficients for the underlying piecewise Linear Time-Invariant (LTI) complex SDE. Consider the interval $[t_i, t_{i+1}]$ with length $\Delta t$. We assume constant coefficients $\alpha_i, \beta_i \in \mathbb{R}$ and an isotropic diffusion intensity $Q_i \in \mathbb{R}_{>0}$. The parameterization outputs the entropy increment $\Delta \mathbf{S}_i = \mathbf{S}(t_{i+1}) - \mathbf{S}(t_i)$ and the phase change $\Delta \boldsymbol{\theta}_i = \int_{t_i}^{t_{i+1}} \beta(\tau)d\tau$.

**1. Recovering Drift Parameters $(\alpha_i, \beta_i)$.** The rotational frequency $\beta_i$ is directly recovered from the mean phase velocity:

$$
\beta_i = \frac{\Delta \boldsymbol{\theta}_i}{\Delta t}.
\tag{71}
$$

To recover the real expansion rate $\alpha_i$, we exploit the relationship between the correlation modulus $|\boldsymbol{\gamma}_{i,i+1}|$ and the entropy. From Theorem 3.3, $|\boldsymbol{\gamma}_{i,i+1}| = \exp(-\Delta \mathbf{S}_i/2)$. Alternatively, from the analytical solution of the LTI system, the modulus

satisfies $|\gamma_{i,i+1}| = e^{\alpha_i \Delta t}\sqrt{\Sigma(t_i)/\Sigma(t_{i+1})}$. ating these two forms and taking the logarithm yields:

$$-\frac{\Delta \mathbf{S}_i}{2} = \alpha_i \Delta t + \frac{1}{2}\ln \Sigma(t_i) - \frac{1}{2}\ln \Sigma(t_{i+1}). \tag{72}$$

Solving for $\alpha_i$, we obtain the drift as a balance between variance growth and entropy injection:

$$\alpha_i = \frac{1}{\Delta t}\left(\frac{1}{2}\Delta \ln \Sigma_i - \frac{1}{2}\Delta \mathbf{S}_i\right), \tag{73}$$

where $\Delta \ln \Sigma_i \triangleq \ln \Sigma(t_{i+1}) - \ln \Sigma(t_i)$.

**2. Recovering Diffusion Intensity ($Q_i$).**   The total variance $\Sigma(t)$ evolves according to the ODE $\dot{\Sigma} = 2\alpha_i \Sigma + Q_i$. The general solution at $t_{i+1}$ is:

$$\Sigma(t_{i+1}) = e^{2\alpha_i \Delta t}\Sigma(t_i) + Q_i \mathcal{E}(\alpha_i), \tag{74}$$

where the integral term is $\mathcal{E}(\alpha_i) = \int_0^{\Delta t} e^{2\alpha_i(\Delta t - \tau)}d\tau = \frac{e^{2\alpha_i \Delta t}-1}{2\alpha_i}$. Using the identity derived from the correlation definition, $e^{2\alpha_i \Delta t}\Sigma(t_i) = |\gamma_{i,i+1}|^2 \Sigma(t_{i+1})$, we substitute this into the variance equation:

$$\Sigma(t_{i+1}) = |\gamma_{i,i+1}|^2 \Sigma(t_{i+1}) + Q_i \mathcal{E}(\alpha_i). \tag{75}$$

Solving for $Q_i$ yields the closed-form reconstruction:

$$Q_i = \frac{\Sigma(t_{i+1})(1 - |\gamma_{i,i+1}|^2)}{\mathcal{E}(\alpha_i)}. \tag{76}$$

Since $\Delta \mathbf{S}_i > 0$ implies $|\gamma_{i,i+1}| < 1$, and $\mathcal{E}(\alpha_i)$ is strictly positive, this guarantees a valid diffusion intensity $Q_i > 0$.

**3. Limit Analysis and Consistency.**   We examine the behavior of the derived parameters in the continuous limit $\Delta t \to 0$. For the diffusion intensity, using the first-order Taylor expansion $1 - |\gamma|^2 = 1 - e^{-\Delta \mathbf{S}_i} = \Delta \mathbf{S}_i + O(\Delta t^2)$ and $\mathcal{E}(\alpha_i) = \Delta t + O(\Delta t^2)$:

$$\lim_{\Delta t \to 0} Q_i = \lim_{\Delta t \to 0} \Sigma(t)\frac{\Delta \mathbf{S}_i}{\Delta t} = \Sigma(t)\dot{\mathbf{S}}(t). \tag{77}$$

This aligns perfectly with the thermodynamic definition of entropy rate $\dot{\mathbf{S}} = Q/\Sigma$. Similarly, taking the limit of (73) recovers the instantaneous drift:

$$\alpha(t) \triangleq \lim_{\Delta t \to 0} \alpha_i = \frac{1}{2}\frac{d}{dt}\ln \Sigma(t) - \frac{1}{2}\dot{\mathbf{S}}(t) = \frac{\dot{\Sigma}(t)}{2\Sigma(t)} - \frac{\dot{\mathbf{S}}(t)}{2}. \tag{78}$$

This equation reveals the explicit physical interpretation of $\alpha(t)$: it is the difference between the relative variance growth rate and the half-rate of entropy injection.

*Remark* A.1 (Differentiability and Implementation).   The existence of the instantaneous drift $\alpha(t)$ in (78) mathematically requires $\Sigma(t) \in C^1$. In our implementation, this is naturally satisfied. Since $\Sigma(t)$ is parameterized by a neural network conditioned on continuous temporal embeddings, we employ smooth activation functions (e.g., Softplus or Swish) for the variance head. This ensures that $\dot{\Sigma}(t)$ is well-defined everywhere, guaranteeing the existence of a valid underlying SDE trajectory without singularities.

## A.8. Derivation of the Marginal ELBO with Flow Refinement

We seek to maximize the marginal Evidence Lower Bound (ELBO) defined on the observations $\mathbf{x}_{\mathcal{T}}$ at discrete time points $\mathcal{T}$. Let $\mathbf{z} = \mathbf{z}_{\mathcal{T}}$ denote the latent variables at these times. The ELBO objective is:

$$\mathcal{L}(\boldsymbol{\theta}, \boldsymbol{\phi}) = \mathbb{E}_{q_{\boldsymbol{\phi}}(\mathbf{z}|\mathbf{x})}[\log p_{\boldsymbol{\theta}}(\mathbf{x}|\mathbf{z})] - D_{\mathrm{KL}}(q_{\boldsymbol{\phi}}(\mathbf{z}|\mathbf{x})\|p_{\boldsymbol{\theta}}(\mathbf{z})), \tag{79}$$

where the first term is the reconstruction likelihood and the second is the KL divergence between the variational posterior and the prior induced by the SDE.

**1. Decomposition via Normalizing Flows.** Our variational posterior $q_\phi(\mathbf{z})$ is constructed by pushing a structured base distribution $q_{\text{base}}(\mathbf{u})$ (with $\mathbf{u} \equiv \mathbf{z}_{\text{base}}$) through a conditional normalizing flow $T_\phi$:

$$\mathbf{z} = T_\phi(\mathbf{u}; \mathbf{h}), \quad \mathbf{u} \sim q_{\text{base}}(\mathbf{u}). \tag{80}$$

By the change of variables formula, the log-density of the transformed variable is:

$$\log q_\phi(\mathbf{z}) = \log q_{\text{base}}(\mathbf{u}) - \log \left| \det \mathbf{J}_{T_\phi}(\mathbf{u}) \right|, \tag{81}$$

where $\mathbf{J}_{T_\phi} = \nabla_{\mathbf{u}} T_\phi$ is the Jacobian of the transformation. Substituting this into the KL divergence term yields a tractable Monte Carlo estimator:

$$
\begin{aligned}
-D_{\text{KL}}(q_\phi \| p_{\boldsymbol{\theta}}) &= \mathbb{E}_{q_\phi(\mathbf{z})}[\log p_{\boldsymbol{\theta}}(\mathbf{z}) - \log q_\phi(\mathbf{z})] \\
&= \mathbb{E}_{q_{\text{base}}(\mathbf{u})} \left[ \log p_{\boldsymbol{\theta}}(T_\phi(\mathbf{u})) - \left( \log q_{\text{base}}(\mathbf{u}) - \log \left| \det \mathbf{J}_{T_\phi}(\mathbf{u}) \right| \right) \right] \\
&= \underbrace{\mathbb{E}_{q_{\text{base}}}[\log p_{\boldsymbol{\theta}}(T_\phi(\mathbf{u}))]}_{\text{Prior Matching (MC)}} + \underbrace{\mathbb{E}_{q_{\text{base}}}[\log \left| \det \mathbf{J}_{T_\phi}(\mathbf{u}) \right|]}_{\text{Flow Volume Expansion}} + \underbrace{\mathcal{H}(q_{\text{base}})}_{\text{Analytic Base Entropy}} .
\end{aligned}
\tag{82}
$$

Here, $\mathcal{H}(q_{\text{base}}) = -\mathbb{E}_{q_{\text{base}}}[\log q_{\text{base}}(\mathbf{u})]$ represents the entropy of the base distribution.

**2. Analytical Entropy of the Base Gauss-Markov distribution.** A critical efficiency advantage of our framework is that the base entropy $\mathcal{H}(q_{\text{base}})$ can be computed analytically in $O(ND)$ time, avoiding the cubic $O(N^3)$ scaling required for general Gaussian process posteriors.

The base distribution $q_{\text{base}}(\mathbf{u})$ factorizes according to the first-order Markov property:

$$q_{\text{base}}(\mathbf{u}_{0:N}) = q(\mathbf{u}_0) \prod_{i=0}^{N-1} q(\mathbf{u}_{i+1} | \mathbf{u}_i). \tag{83}$$

By the distribution rule for differential entropy, the total entropy decomposes into:

$$\mathcal{H}(q_{\text{base}}) = \mathcal{H}(\mathbf{u}_0) + \sum_{i=0}^{N-1} \mathcal{H}(\mathbf{u}_{i+1} | \mathbf{u}_i). \tag{84}$$

We derive the explicit form for the complex-valued case (Theorem 3.3). The conditional distribution is given by $\mathbf{u}_{i+1} | \mathbf{u}_i \sim \mathcal{CN}(\boldsymbol{\mu}_{i+1|i}, \boldsymbol{\Sigma}_{\text{cond}})$, where the diagonal conditional variance is $\boldsymbol{\Sigma}_{\text{cond}} = \boldsymbol{\Sigma}_{i+1} \odot (\mathbf{1} - |\boldsymbol{\gamma}_{i,i+1}|^2)$.

Recall that the differential entropy of a complex Gaussian vector $\mathbf{v} \in \mathbb{C}^D$ with diagonal covariance $\mathbf{V}$ is $\sum_{d=1}^{D} \log(\pi e V_d)$. The entropy terms are thus:

$$\mathcal{H}(\mathbf{u}_0) = \sum_{d=1}^{D} \log(\pi e \Sigma_{0,d}), \tag{85}$$

$$\mathcal{H}(\mathbf{u}_{i+1} | \mathbf{u}_i) = \sum_{d=1}^{D} \log \left( \pi e \Sigma_{i+1,d}(1 - |\gamma_{i,i+1,d}|^2) \right). \tag{86}$$

Summing these components, we obtain the closed-form entropy:

$$\mathcal{H}(q_{\text{base}}) = \underbrace{ND\log(\pi e)}_{\text{Constant}} + \underbrace{\sum_{i=0}^{N} \sum_{d=1}^{D} \log \Sigma_{i,d}}_{\text{Marginal Variances}} + \underbrace{\sum_{i=0}^{N-1} \sum_{d=1}^{D} \log(1 - |\gamma_{i,i+1,d}|^2)}_{\text{Correlation Constraints}}. \tag{87}$$

*Remark* A.2 (Real-Valued Case). For real-valued processes, the constant term becomes $\frac{ND}{2} \log(2\pi e)$ and the log-variance terms are scaled by $\frac{1}{2}$. The structural dependency on $\Sigma$ and $\gamma$, however, remains identical, ensuring the implementation is agnostic to the domain choice (up to constants).

**3. Final Objective.** Combining the analytic entropy with the Monte Carlo estimates for the flow and reconstruction terms, the final optimization objective is:

$$\mathcal{L} \approx \frac{1}{S} \sum_{s=1}^{S} \left( \log p_{\boldsymbol{\theta}}(\mathbf{x}|\mathbf{z}^{(s)}) + \log p_{\boldsymbol{\theta}}(\mathbf{z}^{(s)}) + \log \left| \det \mathbf{J}_{T_{\boldsymbol{\phi}}}(\mathbf{u}^{(s)}) \right| \right) + \mathcal{H}(q_{\text{base}}), \tag{88}$$

where $\mathbf{u}^{(s)} \sim q_{\text{base}}$ and $\mathbf{z}^{(s)} = T_{\boldsymbol{\phi}}(\mathbf{u}^{(s)})$. This objective encourages the flow to expand the probability volume (maximize Jacobian determinant) to capture multimodal uncertainty, while the analytic entropy term acts as a regularizer ensuring the base process maintains a valid Markovian structure.

*Remark* A.3. While evaluating $\log p_{\boldsymbol{\theta}}(\mathbf{z}^{(s)})$ for general nonlinear SDEs still requires integration, it is significantly more efficient than path-based inference because it eliminates the need to sequentially simulate trajectories from the initial state. Instead, we leverage the Markov property to factorize the sequence into a Markov distribution, enabling parallel simulation. In this work, we adopt efficient approximate filtering algorithms (Ansari et al., 2023), which proves sufficient for our experiments. See Section C.4 for details.

# B. Dataset Details and Preprocessing

We evaluate our proposed method on four benchmark datasets spanning medical, meteorological, sensor-based, and synthetic domains. Following established protocols in continuous-time modeling (Becker et al., 2019; Schirmer et al., 2022), we adopt standard preprocessing and splitting schemes to ensure a fair comparison. A summary of the dataset statistics is provided in Table 6.

*Table 6.* Summary of dataset statistics. $N$ denotes the total number of sequences, $T$ the maximum sequence length, and $D$ the input feature dimension.

| Dataset | Type | Task | N (Train/Val/Test) | T | D |
|---|---|---|---|---|---|
| Physionet 2012 | Medical | Inter/Extrapolation | 8,000 | 168 | 37 |
| USHCN | Climate | Inter/Extrapolation | 1,168 | 1461 | 5 |
| Human Activity | Sensor | Classification | 6,554 | 50 | 12 |
| Pendulum | Physics | Regression | 4,000 | 100 | $24^2$ |

## B.1. Physionet Challenge 2012

The Physionet dataset(Silva et al., 2012) [1] consists of multivariate clinical time series derived from 8,000 ICU records. We excluded static descriptors (e.g., age, gender) and retained 37 time-variant physiological features. Timestamps were quantized to the nearest 6 minutes. Each feature was independently normalized to the $[0, 1]$ interval across the training, validation, and test sets. Missing values were zero-encoded and masked during the loss computation to handle sparsity. The dataset is partitioned into 60% training, 20% validation, and 20% testing sets.

## B.2. USHCN Daily Weather

The U.S. Historical Climatology Network (USHCN) dataset (Menne et al., 2016) [2] contains daily meteorological measurements from 1,168 stations recorded between 1990 and 1993. We utilize a subset of 5 observations: precipitation, snowfall, snow depth, maximum temperature, and minimum temperature. To ensure data quality, outliers deviating by more than 4 standard deviations from the trajectory mean were removed. To simulate the irregular sampling characteristic of real-world sensors, the data is randomly subsampled. Features are normalized to $[0, 1]$ per split. The data is divided into 60% training, 20% validation, and 20% testing.

## B.3. Human Activity Recognition

This dataset [3] comprises 6,554 sequences of 3D sensor data collected from five individuals performing various activities (e.g., walking, lying down). The data includes 12 features tracked by sensors attached to the belt, chest, and ankles. Each sequence,

---

[1] Available at https://physionet.org/content/challenge-2012/1.0.0/ under ODC-BY 1.0.

[2] Available at https://data.ess-dive.lbl.gov/view/doi:10.3334/CDIAC/CLI.NDP019. under CC BY 4.0

[3] Available at https://doi.org/10.24432/C57G8X under CC BY 4.0.

originally of length 211, is subsampled to 50 irregularly spaced time points to challenge the model's continuous-time handling capabilities. The objective is to classify the activity into one of seven categories at each time step. We utilize standard splits of 4,194 sequences for training, 1,049 for validation, and 1,311 for testing.

### B.4. Pendulum Simulation

We utilize a synthetic dataset of 4,000 sequences generated from a physical pendulum simulation. The input consists of high-dimensional $24 \times 24$ pixel images, which are corrupted by a correlated noise process to simulate measurement noise (Becker et al., 2019). Each sequence has a length of 50, irregularly sampled from a total duration of $T = 100$. The downstream task is to regress the sine and cosine of the pendulum's angle. The dataset is split into 2,000 training, 1,000 validation, and 1,000 test sequences.

## C. Implementation Details

### C.1. Transformer-based Temporal Encoder for Data Assimilation

To effectively aggregate information from irregular observations and amortize the inference of the variational posterior, we employ a Transformer-based architecture adapted for continuous-time settings, following the design principles in Park et al. (2025). This encoder maps the sequence of discrete observations (and queries) to context-aware embeddings $\mathbf{h}_t$, which condition the subsequent flow dynamics and moment parameterization.

#### C.1.1. CONTINUOUS TEMPORAL EMBEDDING

Unlike standard Transformers that rely on discrete index-based positional encodings, irregular time series require embeddings that respect continuous intervals. We utilize sinusoidal positional encodings $\mathbf{p}(t) \in \mathbb{R}^{D_h}$ defined on the absolute timestamps. For dimension $k \in \{0, \ldots, D_h/2 - 1\}$:

$$\mathbf{p}(t)_{2k} = \sin(t/\tau^{2k/D_h}), \quad \mathbf{p}(t)_{2k+1} = \cos(t/\tau^{2k/D_h}), \tag{89}$$

where $\tau = 10000$ is the wavelength scale. The input to the Transformer at time $t_i$ is the element-wise sum of the learned feature projection and the temporal embedding: $\mathbf{e}_i = \mathbf{W}_{\text{in}}\mathbf{x}_i + \mathbf{p}(t_i)$.

#### C.1.2. TASK-DEPENDENT ATTENTION MASKING

The core mechanism for distinguishing between different inference tasks (e.g., interpolation vs. extrapolation) lies in the rigorous design of the attention mask $\mathbf{M} \in \{0, -\infty\}^{N \times N}$. We feed both observed points $\mathcal{O}$ and target prediction points $\mathcal{P}$ into the network. The attention mechanism operates as:

$$\text{Attention}(\mathbf{Q}, \mathbf{K}, \mathbf{V}) = \text{softmax}\left(\frac{\mathbf{Q}\mathbf{K}^\top}{\sqrt{d_k}} + \mathbf{M}\right)\mathbf{V}. \tag{90}$$

Depending on the task, we apply specific masking strategies to enforce the appropriate information flow:

1. **Causal Mode (Filtering/Forecasting):** For extrapolation tasks where predictions must rely solely on past information, we employ a causal mask. The mask entry $M_{ij} = 0$ if and only if $t_j \leq t_i$ and $j \in \mathcal{O}$; otherwise $M_{ij} = -\infty$. This ensures that the embedding at $t_i$ aggregates information strictly from the history $\mathcal{H}_{t_i} = \{(\mathbf{x}_j, t_j) \mid t_j \leq t_i, j \in \mathcal{O}\}$.

2. **Non-Causal Mode (Smoothing/Interpolation):** For interpolation tasks, the model is permitted to utilize the entire observational context. We employ a non-causal mask where $M_{ij} = 0$ for all valid observations $j \in \mathcal{O}$, allowing bidirectional information flow.

#### C.1.3. QUERY-KEY SEPARATION STRATEGY

Crucially, target points $k \in \mathcal{P}$ are included in the input sequence to generate query vectors $\mathbf{q}_k$ corresponding to their timestamps. However, to prevent information leakage during training (or to reflect the unobserved nature during inference), these points are strictly masked out from the key and value sets. Formally:

$$M_{i,k} = -\infty, \quad \forall i \in \{1, \ldots, N\}, \forall k \in \mathcal{P}. \tag{91}$$

This constraint forces the model to reconstruct the latent state at a target time $t_k$ solely by attending to the valid context defined by the assimilation strategy, effectively implementing a *Query-Key Separation* mechanism.

## C.2. Conditional Normalizing Flows

In this section, we detail the formulation of the three specific layers constituting our conditional flow block: ActNorm, Invertible $1 \times 1$ Convolution, and the Conditional Affine Coupling Layer. These components are composed sequentially to construct the deep generative model.

### C.2.1. ACTNORM (ACTIVATION NORMALIZATION)

To alleviate the training instability often encountered in deep generative models, we employ ActNorm (Kingma & Dhariwal, 2018), which performs an affine transformation using a learnable scale and bias parameter per channel. Let $\mathbf{z} \in \mathbb{R}^{B \times D}$ be the input tensor, where $B$ is the batch size and $D$ is the feature dimension. ActNorm operates independently on each channel $j \in \{1, \ldots, D\}$ via:

$$\mathbf{z}'_{:,j} = \mathbf{z}_{:,j} \odot s_j + b_j, \tag{92}$$

where $s_j$ and $b_j$ are trainable parameters.

**Data-Dependent Initialization.** Crucially, these parameters are initialized using the first minibatch of data. We set $s_j$ and $b_j$ such that the post-actnorm activations have zero mean and unit variance initially. Following this initialization, they are treated as standard trainable variables independent of the data. This mechanism effectively acts as a batch normalization variant tailored for flow models, ensuring stable optimization without introducing stochasticity during the inference pass.

### C.2.2. INVERTIBLE $1 \times 1$ CONVOLUTION

Standard flow architectures (e.g., RealNVP) rely on fixed permutations to ensure information exchange across channel dimensions. Following Kingma & Dhariwal (2018), we replace fixed permutations with a learnable invertible $1 \times 1$ convolution, which generalizes the permutation operation. For an input $\mathbf{z}$ and a learnable weight matrix $\mathbf{W} \in \mathbb{R}^{D \times D}$, the operation is defined as the linear transformation $\mathbf{z}' = \mathbf{z}\mathbf{W}$.

To efficiently compute the log-determinant of the Jacobian required for likelihood evaluation, we parameterize $\mathbf{W}$ via its LU decomposition:

$$\mathbf{W} = \mathbf{PL}(\mathbf{U} + \mathrm{diag}(\mathbf{s})), \tag{93}$$

where $\mathbf{P}$ is a fixed permutation matrix, $\mathbf{L}$ is a lower triangular matrix with ones on the diagonal, $\mathbf{U}$ is an upper triangular matrix with zeros on the diagonal, and $\mathbf{s}$ is a vector. The log-determinant is thus reduced to the sum of the logarithmic diagonal terms:

$$\log|\det(\mathbf{W})| = \sum_{j=1}^{D} \log|s_j|. \tag{94}$$

This parametrization reduces the computational complexity of the determinant from $\mathcal{O}(D^3)$ to $\mathcal{O}(D)$, making the operation scalable to high-dimensional spaces.

### C.2.3. CONDITIONAL AFFINE COUPLING LAYER

The core component of our flow is the conditional affine coupling layer, which we extend to incorporate the continuous temporal context $\mathbf{h}_t$. This design draws inspiration from the conditional blocks in cINN (Ardizzone et al., 2019).

Let $\mathbf{u}$ be the input to the coupling layer. We partition $\mathbf{u}$ along the feature dimension into two halves, $\mathbf{u}_a, \mathbf{u}_b \in \mathbb{R}^{D/2}$. The transformation keeps $\mathbf{u}_a$ invariant while applying an affine transformation to $\mathbf{u}_b$, parameterized by $\mathbf{u}_a$ and the context $\mathbf{h}_t$:

$$\mathbf{u}'_a = \mathbf{u}_a, \tag{95}$$

$$\mathbf{u}'_b = \mathbf{u}_b \odot \exp\left(s_\psi(\mathbf{u}_a, \mathbf{h}_t)\right) + t_\psi(\mathbf{u}_a, \mathbf{h}_t), \tag{96}$$

where $s_\psi(\cdot)$ (scale) and $t_\psi(\cdot)$ (translation) are realized by neural networks. In our implementation, we use a shallow Multi-Layer Perceptron (MLP) with gated activation units. Conditioning on $\mathbf{h}_t$ allows the flow to dynamically adapt the density transformation at each time step, enabling the modeling of non-stationary posterior distributions.

The inverse transformation is analytically efficient:

$$\mathbf{u}_a = \mathbf{u}_a', \tag{97}$$

$$\mathbf{u}_b = (\mathbf{u}_b' - t_\psi(\mathbf{u}_a, \mathbf{h}_t)) \odot \exp\left(-s_\psi(\mathbf{u}_a, \mathbf{h}_t)\right). \tag{98}$$

The final output is the concatenation $\mathbf{u}' = [\mathbf{u}_a', \mathbf{u}_b']$. Since the Jacobian of this transformation is triangular, the log-determinant is simply the sum of the scaling factors:

$$\log\left|\det \frac{\partial \mathbf{u}'}{\partial \mathbf{u}}\right| = \sum \log |\exp(s_\psi(\mathbf{u}_a, \mathbf{h}_t))| = \sum s_\psi(\mathbf{u}_a, \mathbf{h}_t). \tag{99}$$

### C.3. Parallel Sampling via Associative Scan

To achieve logarithmic time complexity $O(\log N)$ for sampling latent trajectories, we reformulate the sequential generation process as a parallel prefix sum problem (associative scan). This section details the mathematical mapping, the proof of associativity, and the specific algorithmic steps used in our implementation.

#### C.3.1. LINEAR RECURRENCE DECOMPOSITION

The generative process in our framework constitutes a first-order linear recurrence relation. For a sequence of length $N$, let $i \in \{0, \dots, N-1\}$. The state transition from $t_i$ to $t_{i+1}$ is given by:

$$\tilde{\mathbf{z}}_{t_{i+1}} = \mathbf{A}_i \tilde{\mathbf{z}}_{t_i} + \mathbf{b}_i + \mathbf{c}_i \odot \boldsymbol{\epsilon}_{i+1}, \quad \boldsymbol{\epsilon}_{i+1} \sim \mathcal{N}(\mathbf{0}, \mathbf{I}), \tag{100}$$

where $\mathbf{A}_i, \mathbf{b}_i, \mathbf{c}_i$ are parameters predicted by the neural network. To apply the parallel scan, we decompose this equation into the standard affine form $\mathbf{z}_{\text{out}} = \mathbf{S} \odot \mathbf{z}_{\text{in}} + \mathbf{V}$. We define two tensors, **Slopes (S)** and **Values (V)**, which serve as the inputs to the scan algorithm.

Since the noise vectors $\boldsymbol{\epsilon}$ are independent and identically distributed (i.i.d.), we can pre-sample the entire noise tensor $\mathbf{E} \in \mathbb{R}^{B \times N \times D}$ and pre-compute the effective coefficients for all time steps in parallel:

- **Slope Tensor** $\mathbf{S}_i = \mathbf{A}_i = \frac{\mathbf{\Sigma}(t_{i+1})}{\mathbf{\Sigma}(t_i)} \odot \boldsymbol{\gamma}_{i,i+1}.$

- **Value Tensor** $\mathbf{V}_i = (\boldsymbol{\mu}(t_{i+1}) - \mathbf{A}_i \boldsymbol{\mu}(t_i)) + \left(\mathbf{\Sigma}(t_{i+1}) \odot \sqrt{1 - \boldsymbol{\gamma}_{i,i+1}^2}\right) \odot \boldsymbol{\epsilon}_{i+1}.$

#### C.3.2. THE ASSOCIATIVE OPERATOR

The core of the parallel scan is an associative binary operator. Let $u_i = (\mathbf{S}_i, \mathbf{V}_i)$ represent the transformation tuple at step $i$. We define the binary operator $\bullet$ that composes two consecutive transformations $u_j$ and $u_i$ (where $j > i$) as:

$$u_j \bullet u_i = (\mathbf{S}_j, \mathbf{V}_j) \bullet (\mathbf{S}_i, \mathbf{V}_i) \triangleq (\mathbf{S}_j \odot \mathbf{S}_i, \quad \mathbf{S}_j \odot \mathbf{V}_i + \mathbf{V}_j). \tag{101}$$

This operator has a clear physical interpretation: the cumulative slope is the product of individual slopes, and the cumulative value is the current value plus the previous value scaled by the current slope.

**Proof of Associativity.** For the scan algorithm to be valid, the operator must be strictly associative. Let $u_k, u_j, u_i$ be three consecutive tuples. We verify that $(u_k \bullet u_j) \bullet u_i = u_k \bullet (u_j \bullet u_i)$.

$$(u_k \bullet u_j) \bullet u_i = (\mathbf{S}_k \mathbf{S}_j, \mathbf{S}_k \mathbf{V}_j + \mathbf{V}_k) \bullet (\mathbf{S}_i, \mathbf{V}_i) \tag{102}$$

$$= (\mathbf{S}_k \mathbf{S}_j \mathbf{S}_i, \quad (\mathbf{S}_k \mathbf{S}_j) \mathbf{V}_i + \mathbf{S}_k \mathbf{V}_j + \mathbf{V}_k). \tag{103}$$

Similarly, grouping the right terms first:

$$u_k \bullet (u_j \bullet u_i) = (\mathbf{S}_k, \mathbf{V}_k) \bullet (\mathbf{S}_j \mathbf{S}_i, \mathbf{S}_j \mathbf{V}_i + \mathbf{V}_j) \tag{104}$$

$$= (\mathbf{S}_k (\mathbf{S}_j \mathbf{S}_i), \quad \mathbf{S}_k (\mathbf{S}_j \mathbf{V}_i + \mathbf{V}_j) + \mathbf{V}_k). \tag{105}$$

The results are identical. This associativity allows the operations to be regrouped into a binary tree structure, enabling the divide-and-conquer strategy.

---

**Algorithm 1** Parallel Scan for SDE Sampling (Work-Efficient)

---

1: **Input:** Sequence of tuples $\mathcal{U} = \{u_0, u_1, \ldots, u_{N-1}\}$ where $u_i = (\mathbf{S}_i, \mathbf{V}_i)$.
2: **Output:** Scanned sequence $\mathcal{U}'$ representing cumulative transitions.
3: // Phase 1: Up-Sweep (Reduction)
4: **for** $d = 0$ **to** $\lceil \log_2 N \rceil - 1$ **do**
5:    **for each** $k$ **in parallel do**
6:       $idx \leftarrow (k+1)2^{d+1} - 1$
7:       $child \leftarrow idx - 2^d$
8:       **if** $idx < N$ **then**
9:          $u_{idx} \leftarrow u_{idx} \bullet u_{child}$                                    ▷ Merge children nodes
10:       **end if**
11:    **end for**
12: **end for**
13: // Phase 2: Down-Sweep (Distribution)
14: $u_{N-1} \leftarrow (\mathbf{1}, \mathbf{0})$                                        ▷ Identity element for root
15: **for** $d = \lceil \log_2 N \rceil - 1$ **down to** $0$ **do**
16:    **for each** $k$ **in parallel do**
17:       $idx \leftarrow (k+1)2^{d+1} - 1$
18:       $left \leftarrow idx - 2^d$
19:       **if** $idx < N$ **then**
20:          $temp \leftarrow u_{left}$
21:          $u_{left} \leftarrow u_{idx}$                               ▷ Pass down accumulator
22:          $u_{idx} \leftarrow u_{idx} \bullet temp$                     ▷ Combine with left child
23:       **end if**
24:    **end for**
25: **end for**
26: // Final State Reconstruction
27: **for** $i = 0$ **to** $N - 1$ **in parallel do**
28:    $\tilde{\mathbf{z}}_{t_{i+1}} = u'_i.\mathbf{S} \odot \tilde{\mathbf{z}}_{t_0} + u'_i.\mathbf{V}$
29: **end for**
30: **Return** Trajectory $\tilde{\mathbf{z}}_{0:N}$

---

### C.3.3. ALGORITHM EXECUTION

We employ the work-efficient parallel scan algorithm (Blelloch scan) (Blelloch, 1990). The procedure involves two passes over a logical binary tree constructed over the time dimension, as visualized below:

The detailed procedure is described in Algorithm 1.

**Complexity and Efficiency.** For a sequence length $N$ and parallel processors $P$, the depth of the calculation graph is $O(\log N)$, compared to $O(N)$ for sequential methods. Furthermore, since our variational family assumes diagonal Gaussians, the matrix multiplication $\mathbf{A}_j \mathbf{A}_i$ in the operator reduces to element-wise products ($\odot$). This reduces the complexity of the binary operator from $O(D^3)$ to $O(D)$, rendering the arithmetic intensity negligible compared to memory access.

### C.4. Approximate Filtering

To estimate the filtering distribution $p(\mathbf{z}_{t_{k+1}} | \mathbf{z}_{t_k}) = \mathcal{N}(\mathbf{z}_{t_{k+1}}; \mathbf{m}_{k+1}, \mathbf{P}_{k+1})$ given the previous estimate at $t_k$, we employ approximate algorithms proposed in NCDSSM (Ansari et al., 2023) and CRU (Schirmer et al., 2022). These methods offer a trade-off between expressivity and computational efficiency, ranging from general non-linear approximations to locally diagonalized analytical solutions.

C.4.1. GENERAL NON-LINEAR APPROXIMATION (NCDSSM-NL)

In the general case where the drift $f_\theta$ is a non-linear neural network, the evolution of the mean and covariance is governed by a set of coupled ordinary differential equations (ODEs).

**1. Coupled ODE System.** The mean evolution follows the drift function directly, while the covariance evolves according to the continuous-time Lyapunov equation based on the local linearization (Jacobian) of the drift:

$$\frac{d\mathbf{m}(t)}{dt} = f_\theta(\mathbf{m}(t), t), \tag{106}$$

$$\frac{d\mathbf{P}(t)}{dt} = \mathbf{F}_z(\mathbf{m}, t)\mathbf{P}(t) + \mathbf{P}(t)\mathbf{F}_z(\mathbf{m}, t)^\top + \mathbf{Q}, \tag{107}$$

where $\mathbf{Q}$ is the diffusion matrix (assuming constant diffusion for simplicity), and $\mathbf{F}_z(\mathbf{m}, t)$ is the Jacobian matrix evaluated at the current mean:

$$\mathbf{F}_z(\mathbf{m}, t) = \left. \frac{\partial f_\theta(\mathbf{z}, t)}{\partial \mathbf{z}} \right|_{\mathbf{z}=\mathbf{m}(t)}. \tag{108}$$

In implementation, this Jacobian is efficiently computed via automatic differentiation (Autograd).

**2. Numerical Integration via Joint State.** Since (107) depends on the instantaneous value of $\mathbf{m}(t)$, the system has no closed-form solution. We solve it numerically by defining a joint state vector $\mathbf{S}(t) \in \mathbb{R}^{M+M^2}$:

$$\mathbf{S}(t) = \begin{bmatrix} \mathbf{m}(t) \\ \text{vec}(\mathbf{P}(t)) \end{bmatrix}. \tag{109}$$

The prediction step involves integrating the joint derivative function from $t_k$ to $t_{k+1}$ using an ODE solver (e.g., Runge-Kutta 4 with step size 0.05):

$$[\mathbf{m}_{k+1}^-, \mathbf{P}_{k+1}^-] \leftarrow \text{ODESolver} \left( \frac{d\mathbf{S}}{dt}, \mathbf{S}_{t_k}, [t_k, t_{k+1}] \right). \tag{110}$$

While expressive, this approach requires $O(M^3)$ or $O(KM^2)$ operations per step due to Jacobian-Matrix products.

C.4.2. LOCALLY LINEAR APPROXIMATIONS (NCDSSM-LL)

To improve efficiency, one can assume the drift is locally linear, i.e., $f(\mathbf{z}) = \mathbf{A}(t)\mathbf{z}$. Under the piecewise constant assumption ($\mathbf{A}(t) \approx \mathbf{A}_{\text{eff}}$ over $[t_k, t_{k+1}]$), the mean has an analytical solution involving the matrix exponential:

$$\mathbf{m}_{k+1}^- = \exp(\mathbf{A}_{\text{eff}}\Delta t)\mathbf{m}_k. \tag{111}$$

However, computing the matrix exponential $\exp(\cdot)$ and solving the integral for the covariance still incurs a cubic complexity $O(M^3)$, which remains a bottleneck for high-dimensional latent spaces.

C.4.3. LOCALLY DIAGONAL LINEAR APPROXIMATION (F-CRU)

To achieve linear scaling $O(M)$, the f-CRU model imposes a structural constraint: the basis matrices sharing a common set of eigenvectors $\mathbf{E}$. The effective transition matrix is diagonalized as $\mathbf{A}(t) = \mathbf{E}\mathbf{\Lambda}(t)\mathbf{E}^\top$, where $\mathbf{\Lambda}(t)$ is a diagonal matrix of eigenvalues.

**1. Closed-Form Mean Prediction.** The matrix exponential simplifies to an element-wise exponential in the eigen-basis:

$$\mathbf{m}_{k+1}^- = \mathbf{E} \left( \exp(\mathbf{\Lambda}_{\text{eff}}\Delta t) \odot (\mathbf{E}^\top \mathbf{m}_k) \right). \tag{112}$$

**2. Closed-Form Covariance Prediction.** The covariance update utilizes the eigen-decomposition to solve the Lyapunov equation analytically. Let $\mathbf{\Sigma}^w = \mathbf{E}^\top \mathbf{P}\mathbf{E}$ be the covariance projected into the eigen-space, and $\mathbf{S}^w = \mathbf{E}^\top \mathbf{Q}\mathbf{E}$ be the projected diffusion. The dynamics decouple element-wise:

$$\frac{d\Sigma_{ij}^w}{dt} = (\lambda_i + \lambda_j)\Sigma_{ij}^w + S_{ij}^w. \tag{113}$$

This scalar ODE admits an exact closed-form solution. For each element $(i, j)$:

$$\Sigma_{ij}^w(t + \Delta t) = \underbrace{\Sigma_{ij}^w(t)e^{(\lambda_i + \lambda_j)\Delta t}}_{\text{Decay/Growth}} + \underbrace{\frac{S_{ij}^w}{\lambda_i + \lambda_j}\left(e^{(\lambda_i + \lambda_j)\Delta t} - 1\right)}_{\text{Noise Integration}}. \qquad (114)$$

Finally, the prior covariance is recovered via projection: $\mathbf{P}_{k+1}^- = \mathbf{E}\boldsymbol{\Sigma}^w(t + \Delta t)\mathbf{E}^\top$. This method reduces the prediction cost to simple matrix multiplications and element-wise operations, significantly accelerating training.

### C.5. Training and Hyperparameters

**Optimization and Regularization.** In all experiments, we employed the Adam optimizer (Kingma & Ba, 2015). To ensure a fair evaluation, we aligned our experimental setup with established baselines: for the Human Activity classification task, we followed the protocol of mTAND (Shukla & Marlin, 2021); for all other datasets, we adhered to the setup used in CRU (Schirmer et al., 2022). We calibrated our model size to ensure the number of parameters is comparable to these baselines. Regarding regularization, we applied a weight decay of $1 \times 10^{-2}$ and gradient clipping for the per-point classification and regression tasks. For interpolation tasks, no weight decay was applied. To prevent overfitting on the Physionet extrapolation task, the training was capped at 350 epochs.

**Resource Usage.** All experiments were conducted on an internal computing cluster. Each experimental run was allocated one NVIDIA GPU (RTX 2080Ti or RTX 3090Ti), 16 CPU cores, and 24GB of RAM.

**Hyperparameter Summary.** Table 7 details the specific hyperparameters used for each dataset, including learning rates, batch sizes, and model dimensions. The parameter counts reflect the final architecture configuration.

*Table 7.* Training Hyperparameters and Model Statistics. $R_d$ denotes the dimension of the latent state (RNN hidden size). The parameter count is reported in Millions (M).

| Dataset | Learning Rate | Epochs | Batch Size | Time Scale | $R_d$ | # Params |
|---|---|---|---|---|---|---|
| Human Activity | $1 \times 10^{-3}$ | 400 | 256 | 1/221 | 288 | 5.55 M |
| Pendulum | $1 \times 10^{-3}$ | 500 | 50 | 0.1 | 20 | 183.7 K |
| USHCN | $1 \times 10^{-3}$ | 500 | 50 | 0.2 | 20 | 184.7 K |
| Physionet | $1 \times 10^{-3}$ | 500 | 100 | 0.3 | 24 | 213.5 K |

### C.6. Sensitivity Analysis of Flow Layers

To evaluate the impact of the flow network's depth on multi-modal density approximation, we conduct a hyperparameter sensitivity study on the number of flow layers $K$, varying it from 0 to 7. Table 8 reports the test MSE for PhysioNet (Interpolation) and USHCN (Extrapolation).

The empirical results show that expanding the flow depth steadily decreases the prediction error, with PhysioNet reaching its peak performance at $K = 6$ and USHCN showing optimal results at $K = 5$. When further increasing the depth to $K = 7$, we observe a slight degradation in performance, which is likely attributed to overfitting or optimization challenges in deeper scaling-translation blocks. To balance the expressivity and parameter efficiency across diverse domains, we select $K = 6$ as the unified default configuration for all downstream tasks.

*Table 8.* Hyperparameter sensitivity of the number of flow layers $K$ on PhysioNet (Interpolation) and USHCN (Extrapolation) tasks in terms of test MSE ($\times 10^{-2}$).

| Task / Layers ($K$) | 0 | 1 | 2 | 3 | 4 | 5 | 6 | 7 |
|---|---|---|---|---|---|---|---|---|
| PhysioNet (Inter.) | 0.063 | 0.060 | 0.056 | 0.053 | 0.052 | 0.051 | **0.048** | 0.056 |
| USHCN (Extra.) | 0.590 | 0.587 | 0.590 | 0.584 | 0.484 | **0.432** | 0.455 | 0.575 |

*Table 9.* Methodology comparison between SDE-VI and existing SDE-based models. **Type** distinguishes between path-based and marginal-based variational approximation. Note that marginal-based methods do not explicitly optimize a posterior SDE during training. †The Kalman posterior drift is analytically determined by the prior and observations. ‡SDE-VI's posterior drift is implied by the parameterized valid marginals. $J$ is adaptively chosen by the differential equation solver and is a function of the stiffness of the differential equation meaning it can change (and possibly explode) while optimizing. In contrast, $R$ is a fixed constant used to control the variance of gradient approximations. In practice, both $J$ and $R$ are typically larger than $T$.

| Method | Prior Drift | Posterior Drift | Type | Cost | GPU Util. |
|---|---|---|---|---|---|
| LatentSDE (Li et al., 2020) | $\mathbf{f}_\theta(t, \mathbf{z})$ | $\mathbf{f}_\theta(t, \mathbf{z}) + \mathbf{u}_\phi(t, \mathbf{z})$ | Path | $\mathcal{O}(J \log J)$ | Sequential |
| CRU (Schirmer et al., 2022) | $\mathbf{F}_\theta \mathbf{z}$ or $\mathbf{F}_\theta^{(i)} \mathbf{z}$ | $\tilde{\mathbf{F}}(t)\mathbf{z} + \tilde{\mathbf{b}}(t)^\dagger$ | Marginal | $\mathcal{O}(T)$ | Sequential |
| ARCTA (Course & Nair, 2023) | $\mathbf{f}_\theta(t, \mathbf{z})$ | $\mathbf{A}_\phi(t)\mathbf{z} + \mathbf{b}_\phi(t)$ | Path | $\mathcal{O}(R)$ | Parallel |
| NCDSSM (Ansari et al., 2023) | $\mathbf{f}_\theta(t, \mathbf{z})$ or $\mathbf{F}_\theta^{(i)} \mathbf{z}$ | $\tilde{\mathbf{F}}(t)\mathbf{z} + \tilde{\mathbf{b}}(t)^\dagger$ | Marginal | $\mathcal{O}(T)$ | Sequential |
| ACSSM (Park et al., 2025) | $\mathbf{A}_\phi^{(i)} \mathbf{z}$ | $\mathbf{A}_\phi^{(i)} \mathbf{z} + \mathbf{b}_\phi^{(i)}$ | Path | $\mathcal{O}(\log T)$ | Parallel |
| **SDE-VI (Ours)** | $\mathbf{f}_\theta(t, \mathbf{z})$ | $\mathbf{g}_\phi(t, \mathbf{z})^\ddagger$ | Marginal | $\mathcal{O}(\log T)$ | Parallel |

# D. Comprehensive Comparisons with Other VI Methods for Neural SDEs

In this section, we provide a theoretical and empirical comparison between SDE-VI and existing variational inference frameworks for time series, focusing on posterior expressivity, inference paradigms, and computational efficiency.

## D.1. Expressivity: Beyond the Kalman Smoothing Limit

To understand the theoretical expressivity of our method, we first analyze the exact posterior structure of Linear Gaussian State Space Models (LGSSMs). Consider a prior defined by a linear time-varying SDE: $d\mathbf{z}_t = \mathbf{F}_t\mathbf{z}_t\, dt + \mathbf{L}_t\, d\mathbf{W}_t$. According to optimal control theory, the exact posterior process given observations $\mathbf{x}_{0:T}$ is governed by a modified SDE (Park et al., 2025):

$$d\mathbf{z}_t = [\mathbf{F}_t\mathbf{z}_t + \mathbf{Q}_t\nabla_\mathbf{z} \log h(t, \mathbf{z}_t)]\, dt + \mathbf{L}_t\, d\mathbf{W}_t, \tag{115}$$

where $h(t, \mathbf{z}_t) = p(\mathbf{x}_{t:T}|\mathbf{z}_t)$ represents the expected future likelihood.

**The Riccati Structure of LGSSMs.** For Linear Gaussian State Space Models, the expected future likelihood $h(t, \mathbf{z}_t) = p(\mathbf{x}_{t:T}|\mathbf{z}_t)$ can be explicitly characterized. Since the dynamics are linear and the likelihoods are Gaussian, $h(t, \mathbf{z}_t)$ takes the form of an unnormalized Gaussian exponent, implying that $\log h$ is a quadratic function of the state:

$$\log h(t, \mathbf{z}_t) = -\frac{1}{2}\mathbf{z}_t^\top \mathbf{\Lambda}_t \mathbf{z}_t + \mathbf{r}_t^\top \mathbf{z}_t + c_t, \tag{116}$$

where $\mathbf{\Lambda}_t \succeq \mathbf{0}$ is the precision matrix (information matrix) and $\mathbf{r}_t$ is the information vector. The posterior drift correction term thus becomes strictly linear:

$$\nabla_\mathbf{z} \log h(t, \mathbf{z}_t) = -\mathbf{\Lambda}_t\mathbf{z}_t + \mathbf{r}_t. \tag{117}$$

The coefficients $\mathbf{\Lambda}_t$ and $\mathbf{r}_t$ are not static; they evolve backward in time according to the continuous-time filtering equations. Specifically, $\mathbf{\Lambda}_t$ satisfies the **Matrix Riccati Differential Equation (MRDE)**:

$$-\frac{d\mathbf{\Lambda}_t}{dt} = \mathbf{F}_t^\top \mathbf{\Lambda}_t + \mathbf{\Lambda}_t\mathbf{F}_t - \mathbf{\Lambda}_t\mathbf{L}_t\mathbf{L}_t^\top\mathbf{\Lambda}_t + \mathbf{H}_t^\top\mathbf{R}_t^{-1}\mathbf{H}_t, \tag{118}$$

and $\mathbf{r}_t$ follows the linear adjoint equation:

$$-\frac{d\mathbf{r}_t}{dt} = (\mathbf{F}_t - \mathbf{L}_t\mathbf{L}_t^\top\mathbf{\Lambda}_t)^\top\mathbf{r}_t + \mathbf{H}_t^\top\mathbf{R}_t^{-1}\mathbf{x}_t, \tag{119}$$

where $\mathbf{H}_t$ and $\mathbf{R}_t$ correspond to the observation matrix and noise covariance (assuming a continuous observation model for theoretical continuity).

**Solving the Riccati ation:** These equations constitute a terminal value problem. They are solved by integrating *backward* in time from $t = T$ to $t = 0$, typically using numerical integrators like the Runge-Kutta methods (RK4), with the terminal condition $\mathbf{\Lambda}_T = \mathbf{0}$ and $\mathbf{r}_T = \mathbf{0}$ (assuming no future information at the end). The necessity of solving this backward ODE

enforces the "Linear Barrier": the posterior drift is strictly constrained to be a linear transformation of $\mathbf{z}_t$, governed by the solution of the Riccati equation.

Consequently, the gradient term $\nabla_\mathbf{z} \log h$ is strictly linear, resulting in a posterior drift that remains a Linear Time-Varying (LTV) function:

$$\mathbf{u}_\phi^*(t, \mathbf{z}_t) = (\mathbf{F}_t - \mathbf{Q}_t \mathbf{\Lambda}_t)\mathbf{z}_t + \mathbf{Q}_t \mathbf{r}_t. \tag{120}$$

This implies that variational methods restricting the family to LGSSMs(Schirmer et al., 2022; Ansari et al., 2023) are theoretically bounded to model unimodal, linear dynamics, limiting their ability to capture complex data distributions.

**SDE-VI's Advantage.** In contrast, although SDE-VI utilizes a linear backbone for efficient sampling, the integration of conditional Normalizing Flows4.3 allows us to warp this base process. As proven in Theorem 4.3, this induces a valid non-linear SDE with state-dependent diffusion $\mathbf{H}_\phi(t, \mathbf{z}_t)$, enabling the model to capture complex, multi-modal posterior dynamics that are mathematically inaccessible to pure Kalman-based approaches.

### D.2. Inference Paradigm: Local Amortization vs. Initial Value Problems

A fundamental limitation of standard path-based VI methods (e.g., Latent SDEs (Li et al., 2020)) is their formulation as an Initial Value Problem (IVP). The stochastic process is strictly determined by the initial state $\mathbf{z}_0$ and the numerical integration of the vector field.

**The Sliding Window Fallacy.** Standard SDE-based methods cannot validly employ sliding window training (e.g., training on a sub-interval $[t_{100}, t_{200}]$ independently). The "start" state $\mathbf{z}_{100}$ follows a complex, evolved distribution $p(\mathbf{z}_{100})$ derived from the Fokker-Planck evolution starting at $t_0$. Arbitrarily re-initializing $\mathbf{z}_{100} \sim \mathcal{N}(\mathbf{0}, \mathbf{I})$ severs the Markov distribution, violates long-term dependencies, and introduces severe distribution shifts.

**Local Marginal Approximation.** SDE-VI circumvents this by parameterizing local marginals directly rather than integrating paths from $t_0$. We leverage the powerful context-extraction capability of Transformers to approximate the global posterior using a local context window $\mathcal{W}_k$:

$$q_\phi(\mathbf{z}_t | \mathcal{D}) \approx q_\phi(\mathbf{z}_t | \mathcal{W}_k), \quad \text{where } |\mathcal{W}_k| \ll |\mathcal{D}|. \tag{121}$$

By decomposing the global objective into local segments $\mathcal{L}(\mathcal{D}) \approx \sum_k \mathcal{L}(\mathcal{W}_k)$, SDE-VI naturally supports sliding window training and inference on infinite streams without violating the underlying SDE constraints.

## E. Stability Analysis of the Sampling Process

In this section, we provide a theoretical guarantee that the proposed recursive sampling mechanism ( (16)) generates statistically bounded trajectories. This stands in sharp contrast to traditional SDE solvers (e.g., Euler-Maruyama), where numerical integration errors can accumulate, leading to finite-time explosion, especially when the drift coefficient $\mathbf{F}_\phi$ is not strictly contractive (i.e., not negative definite).

Our framework avoids this pitfall by decoupling the *distribution prediction* from the *path realization*. Since the marginal parameters $\{\boldsymbol{\mu}_{t_i}, \boldsymbol{\sigma}_{t_i}, \boldsymbol{\gamma}_{t_i, t_{i+1}}\}_{i=0}^N$ are direct outputs of a bounded neural network, the induced stochastic process is stable by construction.

**Proposition E.1** (Probabilistic Boundedness of Sampled Trajectories). *Assume the variational parameters predicted by the encoder are bounded such that $\|\boldsymbol{\mu}_{t_i}\|_\infty \leq M$ and $\epsilon \leq \|\boldsymbol{\sigma}_{t_i}\|_\infty \leq S$ for all $i \in \{0, \dots, N\}$, where $M, S > 0$ are constants. For any failure probability $\delta \in (0, 1)$, there exists a bound $K > 0$ such that the entire sampled trajectory $\{\tilde{\mathbf{z}}_{t_i}\}_{i=0}^N$ remains within the hypercube $[-K, K]^D$ with probability at least $1 - \delta$:*

$$\mathbb{P}\left(\max_{0 \leq i \leq N} \|\tilde{\mathbf{z}}_{t_i}\|_\infty \leq K\right) \geq 1 - \delta. \tag{122}$$

*Proof.* The proof relies on the **Marginal Consistency** property of the Gauss-Markov distribution. Although the trajectory $\{\tilde{\mathbf{z}}_{t_i}\}$ is generated recursively via the conditional transition $q(\mathbf{z}_{t_{i+1}} | \mathbf{z}_{t_i})$, by definition, this sequential sampling procedure yields exact samples from the joint distribution $q(\mathbf{z}_{0:N})$ whose marginals are fully specified by the predicted parameters.

Consequently, for any specific timestamp $t_i$, the sample $\tilde{\mathbf{z}}_{t_i}$ follows the marginal Gaussian distribution exactly:

$$\tilde{\mathbf{z}}_{t_i} \sim \mathcal{N}(\boldsymbol{\mu}_{t_i}, \mathrm{diag}(\boldsymbol{\sigma}_{t_i}^2)). \tag{123}$$

Consider a single dimension $d \in \{1, \ldots, D\}$. The state can be reparameterized as $\tilde{z}_{t_i}^{(d)} = \mu_{t_i}^{(d)} + \sigma_{t_i}^{(d)} \cdot \xi$, where $\xi \sim \mathcal{N}(0, 1)$. Using the standard Gaussian tail inequality (Chernoff bound), for any deviation $\lambda > 0$:

$$\mathbb{P}(|\xi| > \lambda) \leq 2 \exp(-\lambda^2/2). \tag{124}$$

Let $K = M + \lambda S$. The probability that the magnitude of the state exceeds $K$ is bounded by:

$$\begin{aligned}
\mathbb{P}(|\tilde{z}_{t_i}^{(d)}| > K) &= \mathbb{P}(|\mu_{t_i}^{(d)} + \sigma_{t_i}^{(d)}\xi| > M + \lambda S) \\
&\leq \mathbb{P}(|\mu_{t_i}^{(d)}| + |\sigma_{t_i}^{(d)}||\xi| > M + \lambda S) \\
&\leq \mathbb{P}(M + S|\xi| > M + \lambda S) \\
&= \mathbb{P}(|\xi| > \lambda) \leq 2 \exp(-\lambda^2/2).
\end{aligned} \tag{125}$$

To bound the entire trajectory across all $N + 1$ time steps and all $D$ dimensions simultaneously, we apply the Union Bound (Boole's inequality):

$$\begin{aligned}
\mathbb{P}\left(\max_{i,d}|\tilde{z}_{t_i}^{(d)}| > K\right) &\leq \sum_{i=0}^{N}\sum_{d=1}^{D}\mathbb{P}(|\tilde{z}_{t_i}^{(d)}| > K) \\
&\leq (N+1)D \cdot 2 \exp(-\lambda^2/2).
\end{aligned} \tag{126}$$

Setting the right-hand side equal to $\delta$, we solve for the required safety margin $\lambda$:

$$2(N+1)D\exp(-\lambda^2/2) = \delta \implies \lambda = \sqrt{2\ln\frac{2(N+1)D}{\delta}}. \tag{127}$$

Substituting $\lambda$ back into the expression for $K$, we obtain the probabilistic bound:

$$K = M + S\sqrt{2\ln\frac{2(N+1)D}{\delta}}. \tag{128}$$

Thus, with probability at least $1 - \delta$, the generated trajectory is strictly bounded by $K$. $\qquad\square$

**Remark on Numerical Stability vs. Gradient Explosion.** A critical concern in recursive systems implies that terms like $\boldsymbol{\sigma}_{t_{i+1}} \oslash \boldsymbol{\sigma}_{t_i}$ in the transition matrix ( 16) could cause numerical explosion if $\boldsymbol{\sigma}_{t_i} \to 0$ or if the ratio grows exponentially. We clarify that this ratio does not govern the *magnitude* of the state, but rather the *scaling of the update*. Substituting the recursive transition into the centered state dynamics:

$$\begin{aligned}
\tilde{\mathbf{z}}_{t_{i+1}} - \boldsymbol{\mu}_{t_{i+1}} &= \boldsymbol{\gamma}_{i,i+1} \odot \frac{\boldsymbol{\sigma}_{t_{i+1}}}{\boldsymbol{\sigma}_{t_i}} \odot (\tilde{\mathbf{z}}_{t_i} - \boldsymbol{\mu}_{t_i}) + \boldsymbol{\sigma}_{t_{i+1}} \odot \sqrt{1 - \boldsymbol{\gamma}^2} \odot \boldsymbol{\epsilon}_{i+1} \\
&= \boldsymbol{\gamma}_{i,i+1} \odot \frac{\boldsymbol{\sigma}_{t_{i+1}}}{\boldsymbol{\sigma}_{t_i}} \odot (\boldsymbol{\sigma}_{t_i} \odot \hat{\boldsymbol{\xi}}_i) + \ldots
\end{aligned} \tag{129}$$

where $\hat{\boldsymbol{\xi}}_i$ represents the *standardized residual* of the previous state. Mathematically, the denominator $\boldsymbol{\sigma}_{t_i}$ cancels with the implicit scale of the previous sample $\tilde{\mathbf{z}}_{t_i}$. Consequently, the magnitude of the new state is proportional to its own marginal scale $\boldsymbol{\sigma}_{t_{i+1}}$, not the ratio. In practice, since $\boldsymbol{\sigma}_t$ is the output of a neural network (typically passed through a Softplus activation plus a small $\epsilon$), it is bounded away from zero. This guarantees that the explicit calculation of the ratio remains numerically stable, enabling our model to learn flexible variance scaling (expressivity) without risking the instability inherent in integrating unbounded SDE drifts.

