# OpenReview forum: "Generative Modeling of Irregular Time Series via SDE-Induced Continuous-Discrete Variational Inference"
_ICML.cc/2026/Conference — ICML 2026 spotlight_

### Official Review · Reviewer_XRMj · 2026-03-05

**Soundness:** 3
**Presentation:** 3
**Significance:** 3
**Originality:** 3
**Overall Recommendation:** 5
**Confidence:** 3

**Summary:**

The paper proposes SDE-VI, a variational inference framework for irregular time series that shifts from path-based ELBOs to a marginal ELBO defined on the joint distribution of latent states at the observed timestamps. The key idea is to parameterize the variational posterior as a Gauss–Markov joint distribution over the discrete latent states, while enforcing that this discrete-time distribution is SDE-induced. Specifically, the posterior is parameterized by $(\{\mu_{t_i}, \sigma_{t_i}, \gamma_{t_i,t_{i+1}}\})$, and the paper derives necessary-and-sufficient conditions for these parameters to correspond to a valid underlying LTV-SDE. To increase expressivity beyond diagonal-linear Gaussian posteriors, the authors refine samples at each timestamp with conditional normalizing flows and prove the resulting distribution is induced by a nonlinear SDE. They further generalize the framework to a complex-valued setting to encode oscillatory dynamics. Experimental results show the effectiveness of the proposed method.

**Compliance With Llm Reviewing Policy:**

Affirmed.

**Key Questions For Authors:**

Please see Weaknesses.

**Limitations:**

None.

**Strengths And Weaknesses:**

Strengths:

- Existing Neural SDEs often pay a high computational price for path-based VI or restrict the posterior heavily (e.g., linear drifts), and the paper directly targets this bottleneck.
- The “marginal-based ELBO” viewpoint is conceptually appealing for irregular data: observations only occur at discrete timestamps, so optimizing a discrete-time joint distribution can be the right object.
- The effectiveness of the proposed SDE-VI is evaluated on multiple benchmarks.


Weaknesses:

- The training objective uses $E_{q_\phi}[\log p_\theta(\mathcal{X},\mathcal{Z})] + \mathcal{H}(q_\phi(\mathcal{Z}))$, but the main text does not clearly specify how the joint density $p_\theta(\mathcal{X}, \mathcal{Z})$ is computed in practice.
- The experiments are mostly conducted on a subset of the datasets, without using all four datasets. Could you clarify the reason for this, or would it be possible to include additional experiments?
- There is no comparison of the inference runtime with baseline methods.

If the authors can clarify these points, I would be willing to increase the score.

---

> ### Author Rebuttal · Authors · 2026-03-31
>
> We sincerely thank you for your positive review and suggestions. We will address your questions point by point below.
>
> **Q1 How is the joint density $p_\theta(\mathcal{X}, \mathcal{Z})$ is computed in practice?**
>
> Due to space constraints, the computation of $p_\theta(\mathcal{X}, \mathcal{Z})$ is detailed in Appendix A.8 and C.4. We are glad to provide a more complete explanation here.
>
> The joint density $p_\theta(\mathcal{X}, \mathcal{Z})$ factorizes into the prior dynamics $p_\theta(\mathcal{Z})$ and the emission likelihood $p_\theta(\mathcal{X}|\mathcal{Z})$. Because the observations are conditionally independent, the emission likelihood factorizes over individual time steps as $p_\theta(\mathcal{X}|\mathcal{Z}) = \prod_{i=0}^N p_\theta(x_{t_i}|z_{t_i})$. We typically compute each local likelihood $p_\theta(x_{t_i}|z_{t_i})$ as a Gaussian distribution $\mathcal{N}(x_{t_i}; e_\theta(z_{t_i}), \sigma^2 I)$. Here, $e_\theta(\cdot)$ is a neural network that maps the latent state into the observation space. Therefore, we evaluate this term directly at each discrete observation timestamp $t_i$ using the sampled latent state $\tilde{z}_{t_i}$.
>
> To compute the joint prior density $p_\theta(\mathcal{Z})$, we use the SDE's Markov property to factorize it as $p_\theta(\mathcal{Z}) = p_\theta(z_{t_0}) \prod_{i=0}^{N-1} p_\theta(z_{t_{i+1}}|z_{t_i})$. The initial state $p_\theta(z_{t_0})$ is parameterized as a learnable Gaussian $\mathcal{N}(\mu_{t_0}, \Sigma_{t_0})$. Therefore, the only remaining step is to compute the transition probability $p_\theta(z_{t_{i+1}}|z_{t_i})$ *in parallel*.
>
> The transition probability $p_\theta(z_{t_{i+1}}|z_{t_i})$ is governed by the Fokker-Planck equation (FPE):
> $$\frac{\partial p}{\partial t} = -\sum_{j=1}^{D} \frac{\partial}{\partial z_j} [f_\theta^j(t, z) p] + \frac{1}{2} \sum_{j,k=1}^{D} \frac{\partial^2}{\partial z_j \partial z_k} [(L_\theta L_\theta^\top)_{jk} p].$$
>
> For a linear drift $f_\theta(t, z_{t}) = F_\theta(t) z_{t} + b_\theta(t)$ and time-dependent diffusion $L_\theta(t, z_{t}) = L_\theta(t)$, the transition probability has an exact Gaussian analytical solution: $p_\theta(z_{t_{i+1}}|z_{t_i}) = \mathcal{N}(z_{t_{i+1}}; \mu_{t_{i+1}|t_i}, \Sigma_{t_{i+1}|t_i})$. Specifically, the conditional mean $\mu_{t_{i+1}|t_i}$ and covariance $\Sigma_{t_{i+1}|t_i}$ can be analytically solved using the Lyapunov differential equations over the interval $[t_i, t_{i+1}]$ as presented in Equation (12). Therefore, we can efficiently compute these exact Gaussian densities in parallel.
>
> For a non-linear drift $f_\theta(t, z_t)$ or a state-dependent diffusion $L_\theta(t, z_t)$, the FPE has no analytical solution, and solving it numerically is generally computationally intractable. A common solution is to approximate the transition density as a Gaussian $\mathcal{N}(\mu_{t_{i+1}}, \Sigma_{t_{i+1}})$, which simplifies the task by tracking only the evolution of the first two moments: the mean $\mu_t$ and covariance $\Sigma_t$. Specifically, initialized with $\mu_{t_i} = z_{t_i}$ and $\Sigma_{t_i} = 0$, the parameters are obtained by numerically integrating the following coupled ODEs over $[t_i, t_{i+1}]$:
> $$\frac{d\mu_t}{dt} = f_\theta(t, \mu_t), \quad \frac{d\Sigma_t}{dt} = J_{f_\theta} \Sigma_t + \Sigma_t J_{f_\theta}^\top +  L_\theta(t, \mu_t) L_\theta(t, \mu_t)^\top,$$
> where $J_{f_\theta} = \nabla_z f_\theta(t, z)|_{z=\mu_t}$ denotes the local Jacobian of the drift. This system can be efficiently solved using general-purpose numerical ODE solvers (e.g., RK4).
>
> **Q2 Why the experiments are mostly conducted on a subset of the datasets, without using all four datasets? Would it be possible to include additional experiments?**
>
> Due to the large scale of the datasets and the extensive training time required(see Table 1), previous studies [1-3] conducted experiments on specific subsets. For a fair comparison, we strictly followed the same subset selection and settings as previous works. This allowed us to compare our results directly against their reported best performance.
>
> **Q3: There is no comparison of the inference runtime with baseline methods.**
>
> The training time of our method and its comparison with relevant Neural SDE models are presented in the Appendix D.3 (see Table 7). For the convenience of your assessment, we now present it, along with the inference time, in Table 1 below.
>
>
> ---
> *Table 1. Runtime comparison (seconds/epoch)*
> |Model|USHCN Train./Infer.|PhysioNet Train./Infer.|
> |-|-|-|
> |RKN-$\Delta_t$|32.1/9.7|114.9/41.9|
> |GRU-D|99.6/35.2|2457/1005.3|
> |Latent-ODE|37.5/13.6|560/240.1|
> |ODE-RNN|27.6/9.5|295.4/116.0|
> |GRU-ODE-B|132.6/36.3|527.7/167.9|
> |CRU|41.6/11.5|302.7/98.2|
> |ACSSM|21.6/5.7|78.6/23.6|
> |SDE-VI|**9.3/3.3**|**40.2/16.8**|
>
> [1] Modeling Irregular Time Series with Continuous Recurrent Units
>
> [2] Neural Continuous-Discrete State Space Models for Irregularly-Sampled Time Series
>
> [3] Amortized Control of Continuous State Space Feynman-Kac Model for Irregular Time Series

---

> > ### Author Rebuttal · Reviewer_XRMj · 2026-04-01
> >
> > Thank you for the detailed reply. All my concerns have been well addressed and my score has increased to 5.

---

### Official Review · Reviewer_9nje · 2026-03-08

**Soundness:** 3
**Presentation:** 3
**Significance:** 3
**Originality:** 2
**Overall Recommendation:** 4
**Confidence:** 3

**Summary:**

The paper proposes SDE-VI, which shifts from path-based to marginal-based variational inference for irregular time series. Instead of learning the full posterior SDE path, they learn distribution parameters at discrete timestamps and enforce that these correspond to a valid SDE. They extend the base linear Gaussian framework with Normalizing Flows (for nonlinearity) and complex-valued inference (for oscillations). They test on four benchmarks and report improvements over existing Neural SDE baselines.

**Compliance With Llm Reviewing Policy:**

Affirmed.

**Final Justification:**

The concession on Theorem 4.3 is appreciated. I encourage the authors to (1) tighten the continuous differentiability argument for Q3 and (2) briefly discuss why SDE-VI's ELBO values are much higher than ACSSM's despite better task performance. I weigh the strengths (particularly the convincing empirical results with advanced priors) as outweighing the remaining concerns about presentation and minor theoretical loose ends. The rebuttal changed my evaluation upward.

**Key Questions For Authors:**

1. What filtering algorithm was used for the prior in the main experiments? Can you show results with each variant?

2. Can you provide a proper explanation for the USHCN interpolation gap? Have you tried adding smoothness regularization?

3. Theorem 4.2 requires continuously differentiable μ(t) and σ(t), but these are Transformer outputs at discrete points. How is this condition actually satisfied in practice?

4. How sensitive is performance to the number of flow layers K? Did you tune this per dataset?

5.Can you report NLL values? Since you optimize an ELBO, showing how tight the bound is would be informative.

**Limitations:**

1.	The prior model is underspecified. Which filtering algorithm was used in the experiments? This is a critical implementation detail that is buried in the appendix and never clearly stated for the main experiments.

2.	Four datasets, all relatively small-scale. No comparison with recent methods like Oh et al. (2024) which is cited but not benchmarked. No diffusion-based time series models. No large-scale or real industrial data. For the claims being made, I expected more.

**Strengths And Weaknesses:**

## Strengths

1.	The core idea is well-motivated. We only observe data at discrete timestamps, so why bother with the full path measure? The marginal-based ELBO being strictly tighter than the path-based one is clear.

2.	The computational speedups are substantial. 4–7× faster than CRU and about 2× faster than ACSSM, with the flow and complex extensions adding almost no overhead.

## Weaknesses

1.	The improvements on several tasks are not statistically significant. On Human Activity, 91.9±0.7 vs 91.4±0.4 for ACSSM; the error bars overlap. On Pendulum, 2.68±0.17 vs 2.98±0.30  again, marginal. The paper claims state-of-the-art but many of these gains could just be noise from three runs.

2.	The USHCN interpolation result is concerning. SDE-VI gets 0.044 while ACSSM gets 0.006 and even RKN-Δt gets 0.009. That is nearly an order of magnitude worse. The authors wave this away by saying convex-combination models have a structural prior for smooth data, but if your model is supposedly more expressive, it should be able to learn smooth dynamics too. This needs a real explanation.

3.	Theorem 4.3 is presented as a contribution but it is essentially a standard application of Ito’s lemma to a diffeomorphism of an SDE. This is textbook material. I appreciate the formal statement, but calling it a novel theorem is a stretch. The real engineering contribution is the Glow architecture with time-conditioned coupling; that part is fine but should be presented as such.

---

> ### Author Rebuttal · Authors · 2026-03-31
>
> Thanks for your comments.
>
> **W1: The improvements on Human Activity and Pendulum are not statistically significant. The error bars overlap.**
>
> We agree that the performance bars of several datasets overlap. However, bar overlapping is very common in performance comparison. For example, the bars reported in baselines ACSSM (2025) and mTAND (2021) overlap, which are 91.4±0.4 vs. 91.1±0.2 on Human Activity and 2.98±0.30 vs. 3.20±0.60 on Pendulum.
>
> Our method's gains are not large because we used a very simple prior (i.e., the locally linear prior) in all of our experiments. To achieve more remarkable improvements, we experimented with two additional priors within our framework:
>
> - First, a convex-combination locally diagonal linear prior, named SDEVI-C. Instead of a hard switch at observation points with arbitrary linear drift, such prior smoothly interpolate between linear dynamics via convex-combinations of several basis matrices.
> - Second, unlike other methods which are restricted to linear priors, our framework supports non-linear prior models $dz_t = f_\theta(t, z_t)dt + L_\theta(t, z_t)dW_t$. Thus, we implemented a nonlinear model(SDEVI-NL).
>
> Results in Table 1 below show the effectiveness and flexibility of our proposed inference framework, especially when it is used together with advanced priors.
>
> Table 1: Performance of advanced priors across benchmarks.
> |Model|Act.Acc|Pend.MSE|USHCN In.|USHCN Ex.|
> |-|-|-|-|-|
> |mTAND|91.1±0.2|3.20±0.60|1.766±0.009|2.360±0.038|
> |NCDSSM|88.5±1.7|2.96±0.16|0.052±0.005|0.906±0.019|
> |ACSSM|91.4±0.4|2.98±0.30|0.006±0.001|0.941±0.014|
> |SDEVI(Ours)|91.9±0.7|2.68±0.17|0.044±0.012|0.455±0.003|
> |SDEVI-C(Ours)|92.7±0.7|2.54±0.12|0.013±0.004|0.431±0.006|
> |SDEVI-NL(Ours)|**93.3±0.4**|**2.30±0.27**|**0.004±0.001**|**0.410±0.010**|
>
> **W2&Q2: Can you provide a proper explanation for the USHCN interpolation gap?**
>
> After investigation, we find that the USHCN dataset exhibits much higher stationarity than other datasets(5x higher volatility compared to PhysioNet), which makes the use of a smooth prior particularly crucial for USHCN. We evaluated SDEVI on the USHCN dataset using the convex-combinations prior(SDEVI-C) and the general non-linear prior(SDEVI-NL). As shown in Table 1, the performance improves significantly.
>
> **W3: The real contribution in Sec. 4.3 is the flow part rather than the Theorem 4.3**
>
> We agree that given the conclusion of Theorem 4.2, Theorem 4.3 is a direct result of applying Itô's lemma. We will reflect this point in our final version.
>
> **Q1&L1: What prior and filtering algorithm was used in the main experiments? Can you show results with each variant?**
>
> For simplicity, the main experiments used a locally diagonal linear prior: $dz = F_\theta(z_{t_i})z dt + dW$, where $p_\theta(z_{t_{i+1}}|z_{t_i})$ can be analytically solved via Eq.(12). As introduced in W1, we also implemented two variants  SDEVI-C and SDEVI-NL(details in App C.4). Table 1 shows these advanced priors consistently improve performance. We will explicitly detail this in the revision.
>
>
> **Q3: Theorem 4.2 requires continuously differentiable μ(t) and σ(t). How is this condition satisfied?**
>
> For an arbitrary timestamp $t$, it is first embedded into a sinusoidal temporal embedding $p(t)$, which is essentially a continuous function of timestamp $t$. Hence, the entire network acts as a continuously differentiable function over time $f: t \mapsto (\mu(t), \sigma(t))$.
>
> **Q4: Sensitivity to flow layers K? Did you tune this per dataset?**
>
> We set K=6 for all tasks. We tuned K from 0 to 7 and found performance steadily improved up to K=6 (e.g., PhysioNet MSE drops from 0.063 at K=0 to 0.048 at K=6, USHCN drops from 0.590 to 0.455) and degrades slightly at K=7.
>
> **Q5: Can you report NLL values?**
>
> True NLL estimation is generally intractable for irregular time series. To show a consistently tighter lower bound, we compare our ELBO against other variational methods using a linear prior:
>
> *Table 2 ELBO results across all benchmarks*
> |Model|Act.|Pend.|USH. In.|USH. Ex.|Phy. In.|Phy. Ex.|
> |-|-|-|-|-|-|-|
> |NCDSSM|-5.8|-4.4|1.1|1.0|-0.5|-3.6|
> |ACSSM|-20.9|-41.4|-27.1|-67.9|-30.3|-25.6|
> |SDEVI|**-5.0**|**-3.2**|**2.2**|**1.8**|**0.2**|**-2.2**|
>
> **L2-1 & L2-2: Missing baselines (Oh et al., 2024 and Diffusion models) and include recent baselines**
>
> We aim at generative modeling via latent SDEs. Oh et al. (2024) use SDEs in a discriminative framework without posterior inference. Diffusion models learn distributions over fixed timestamps, lacking the continuous-time Markovian states needed for arbitrary time queries (interpolation/extrapolation) or classification. We added another recent baseline, NCDSSM (2023), to Tables 1-2.
>
>
> **L2-3: Dataset scale**
>
> These four datasets are standard benchmarks and among the largest real-world irregular datasets available (e.g., PhysioNet: 1.6M observations; Human Activity: 1.25M). We use them to ensure direct and fair comparisons against the best reported results.

---

> > ### Author Rebuttal · Reviewer_9nje · 2026-04-03
> >
> > The concession on Theorem 4.3 is appreciated. Two minor residual concerns: (1) the justification for continuous differentiability of μ(t) and σ(t) (Q3) remains somewhat informal (we encourage the authors to tighten this argument in the revision); (2) the ELBO comparison in Table 2 shows a striking gap between SDE-VI and ACSSM (e.g., -5.0 vs -20.9 on Activity) that deserves brief discussion, since higher ELBO with better downstream performance is worth explaining to readers. I will raise my score to 4.

---

> > > ### Author Response · Authors · 2026-04-03
> > >
> > > **Q1: The justification for continuous differentiability of μ(t) and σ(t) (Q3) remains somewhat informal, we encourage the authors to tighten this argument in the revision.**
> > >
> > > We appreciate your suggestion to tighten this argument. The continuous differentiability of $\mu(t)$ and $\sigma(t)$ is guaranteed by the composition of continuously differentiable functions within our network architecture. Specifically, for an arbitrary temporal input $t \in \mathbb{R}$:
> > >
> > > - **Temporal Embedding**: The timestamp $t \in \mathbb{R}$ is initially mapped via sinusoidal temporal embeddings $p(t)$, which are smooth functions ($C^\infty$).
> > > - **Transformer Blocks:** The attention mechanisms and linear projections rely on matrix multiplications and the Softmax function. The Feed-Forward Networks employ smooth activation functions (e.g., Swish or GELU), ensuring all intermediate transformations are $C^1$ or higher.
> > > - **Output Heads:** The parameter heads utilize smooth functions. For instance, the variance head employs a Softplus activation to ensure global positivity, which is a smooth function ($C^\infty$).
> > >
> > > By the chain rule, the entire network acts as a continuously differentiable function over time, $f: t \mapsto (\mu(t), \sigma(t))$, satisfying the conditions required by Theorem 4.2. We provided a brief note regarding this in Appendix A.7 (Remark A.1) and will incorporate this tightened formalization in the final version of the manuscript.
> > >
> > > **Q2: Why does SDEVI show a striking ELBO gap over ACSSM (e.g., -5.0 vs -20.9 on Activity)?**
> > >
> > > The significant ELBO performance gap between ACSSM and SDEVI primarily stems from the massive, irreducible KL divergence penalty $D_{\text{KL}}(\mathbb{Q}_ {\mathbf{\phi}} || \mathbb{P}_\theta)$ caused by ACSSM's overly restrictive structural assumptions.
> > >
> > > Optimizing the ELBO requires the variational posterior drift to match the true posterior drift  $f_\theta(t, z_t) + L_\theta L_\theta^\top \nabla_z \log h(t, z_t)$ as shown in Equation (5). The control term $L_\theta L_\theta^\top \nabla_z \log h(t, z_t)$ incorporates future observation information, where $h(t, z_t) = \mathbb{E}_ {\mathbb{P}_ \theta}[\prod_{j: t_j > t} p_\theta(x_{t_j} | z_{t_j}) | z_t]$ represents the expected future likelihood. Because the emission likelihood $p_\theta(x_{t_j}|z_{t_j})$ is parameterized by non-linear neural networks, this gradient control term is intrinsically highly non-linear.
> > >
> > > However, ACSSM restricts both its prior SDE and approximate posterior SDE to locally Linear Time-Invariant (LTI) systems between adjacent observations. Specifically, it sets the prior drift to $A_\phi^{(i)}z$ and the variational posterior drift to $A_\phi^{(i)}z + b_\phi^{(i)}$. By enforcing this shared linear dynamic, the formulation forces the simple, piecewise constant vector $b_\phi^{(i)}$ to solely approximate the highly non-linear control term $L_\theta L_\theta^\top \nabla_z \log h(t, z_t)$. This results in a massive, irreducible KL penalty $D_{\text{KL}}(\mathbb{Q}_ {\mathbf{\phi}} || \mathbb{P}_\theta)$ that significantly degrades the overall ELBO. Instead, SDEVI imposes minimal structural assumptions and provides a highly expressive posterior, leading to a much tighter lower bound.

---

### Official Review · Reviewer_hJRg · 2026-03-09

**Soundness:** 4
**Presentation:** 2
**Significance:** 3
**Originality:** 4
**Overall Recommendation:** 5
**Confidence:** 3

**Summary:**

The paper considers modelling irregular time series, extending previous work using variational inference of a latent Gauss-Markov SDE. Rather than learning the SDE drift and diffusion, the authors show it is equivalent to directly learn functions controlling various moments of hidden states at observation times. The authors also increase the flexibility of the approximation (1) by adding normalising flow layers (2) using a complex valued underlying SDE to allow periodic behaviour.

**Compliance With Llm Reviewing Policy:**

Affirmed.

**Final Justification:**

The rebuttal addressed all my concerns -- which related to presentation of some material in the paper -- so I've increased my score from 4 (weak accept) to 5 (accept). I thought the other aspects of the paper were also strong, as in my original review.

**Key Questions For Authors:**

Can you address the numbered points in "main weaknesses", especially 1-2. This would let me increase my score.

**Limitations:**

Yes

**Strengths And Weaknesses:**

The four aspects mentioned in the review form are highlighted in bold.

## Strengths

The main idea of modelling moments seems novel and very powerful (**originality**), and it's well supported by a theoretical result (**soundness**). The other methodological extensions also seem like natural and good ideas.

The experimental results on challenging problems are strong (**significance**), and the ablation study is great at showing the effects of the different parts of the methodology.

The paper is generally well written and easy to follow (**presentation**).

## Main weaknesses

1. I found the section on "Inherent Scalability" hard to follow (**presentation**):
  - Is this implemented in the examples or not?
  - Are the windows used for all of (1) the transformer (2) the posterior and (3) the variational approximation?
  - The previous paragraph says a parallel scan algorithm is used for long time series. Does this conflict with using a window approach?
  - "This strategy naturally supports infinite streams" Do you mean a streaming data setting? Can you expand on how this would work?
2. The inequality in (8) is written as non-strict - can you clarify if this is correct? The surrounding text suggests it is meant to be strict: "is a superior objective", "the following inequality holds strictly". Appendix A says it will prove the non-strict inequality (line 617), but then seems to prove the strict equality (lines 638, 657).
3. In line 283 (right column), why assume these SDEs are independent? This seems different to what was done in the non-complex setting.

### Minor weaknesses

* It would be good to use a grammar checker to fix various typos.
* Line 242 (right column). "the transformed process $z^{(0)}_{[0,T]}$..." If I understand correctly, I think the superscript should be removed, as it would denote the base state.
* It would be good to mention in the main paper that there's an appendix on computationally efficiency. I almost included the lack of this as a major weakness.

---

> ### Author Rebuttal · Authors · 2026-03-31
>
> We sincerely thank you for your positive review and suggestions. We will address your questions point by point below.
>
> **W1-1: Is "Inherent Scalability" implemented in the examples or not?**
>
> In our experiments, we implemented the parallel scan algorithm, but didn't implement the window-based acceleration.
>
> The parallel scan is a mature algorithm broadly adopted by modern linear state space models (e.g., Mamba) to accelerate sequential state transitions. In our framework, we apply it to parallelize the recursive sampling of latent trajectories, reducing the time complexity from sequential $O(N)$ to parallel $O(\log N)$ without any performance loss. As shown in our efficiency experiments, this implementation provides a 2~8x speedup over current state-of-the-art Neural SDEs while achieving the best overall results.
>
> However, the window-based acceleration maintains near-optimal performance only if the local window is long enough to provide sufficient context, such that $q_\phi(z|\mathcal{W}_k) \approx p(z|\mathcal{D})$. After investigating extrapolation tasks, we found that current benchmarks still require the full available history to achieve optimal performance. Thus, we follow prior works by utilizing the full history during inference to ensure a fair comparison. Consequently, we presented the "Inherent Scalability" section primarily as a brief conceptual framework at the end of our methodology for future ultra-long sequence applications.
>
> **W1-2: Are the windows used for all of (1) the transformer (2) the posterior and (3) the variational approximation?**
>
> As clarified in W1-1, the window approach is not employed in our current experiments. However, when the sequence is long enough to apply the window approach, the windows are used for all three components: (1) the Transformer encoder performs data assimilation on the local window $\mathcal{W}_ k$ to extract context-rich embeddings; (2) these embeddings are then used to parameterize the variational approximation $q_\phi(z|\mathcal{W}_k)$; and (2) this approximation is used to approximate the true posterior $p(z|\mathcal{D})$ by assuming that the local context provided by $\mathcal{W}_k$ is sufficient to capture the essential dynamics of the latent state $z$.
>
> **W1-3: Does parallel scan conflict with using a window approach?**
>
> The parallel scan and window approaches are complementary. The window strategy partitions an ultra-long sequence into tractable, non-overlapping windows, while the parallel scan computes state transitions for efficient parallel sampling within each window. However, complex tasks still demand extended window sizes to capture sufficient information for inference. Therefore, the parallel scan is necessary to maintain computational efficiency when these local windows must be large to support accurate inference.
>
> **W1-4: "This strategy naturally supports infinite streams" Do you mean a streaming data setting? Can you expand on how this would work?**
>
> We apologize for the confusion we introduced. "Infinite streams" does not refer to a streaming data setting. We intended to emphasize that our marginal-based framework is inherently scalable for ultra-long sequences. We will clarify this in the revised manuscript.
>
> **W2: Can you clarify whether the inequality in (8)  holds strictly or non-strictly?**
>
> Eq.(8) is a **non-strict** inequality ($\ge$) as equality holds only if the approximate posterior SDE perfectly matches the true posterior. While our objective is "superior" in optimization terms, the word "strictly" is indeed imprecise, and we have removed it in the revised manuscript. We apologize for the confusion and thank you again for your detailed and careful review.
>
> In Appendix A, we present two distinct proof methods for the general non-strict inequality $\mathcal{L}_ {marg} \ge \mathcal{L}_{path}$. Following these proofs, we further demonstrate that under the specific setting of recent baselines where the approximate posterior is restricted to linear, the inequality becomes **strict** ($>$). This is because the linear variational bridge can never perfectly match the non-linear true posterior bridge, making the gap term $\mathbb{E}_q[D _{KL}(\mathbb{Q}(\cdot|\mathcal{Z}) || \mathbb{P}^*(\cdot|\mathcal{Z}))]>0$.
>
> **W3: Why assume complex SDEs are independent? This seems different to what was done in the non-complex setting.**
>
> This is consistent with the real setting (Eq. 6), where diagonal drift $F_{\phi}$ and diffusion $L_{\phi}$ enforce dimensional independence to reduce complexity from $O(D^3)$ to $O(D)$. We extend this to $D/2$ independent complex SDEs to model oscillatory dynamics. While SDEs are independent for efficient parallel scanning, the subsequent flow layers (convolutional) explicitly couple these dimensions, ensuring an expressive posterior with minimal overhead.
>
> **Minor Weaknesses** All typos and the notation error have been corrected. We will include the efficiency experiment in the main text in our final version.

---

> > ### Author Rebuttal · Reviewer_hJRg · 2026-04-02
> >
> > Thanks for the helpful rebuttal comments. This resolves all my concerns. (When writing my Q3 I'd missed the paper's statement that the diffusion matrix was diagonal in the non-complex setting.) I'll increase my score from 4 (weak accept) to 5 (accept).

---

### Official Review · Reviewer_8Ph7 · 2026-03-12

**Soundness:** 3
**Presentation:** 3
**Significance:** 3
**Originality:** 3
**Overall Recommendation:** 5
**Confidence:** 3

**Summary:**

This paper proposes a really fresh perspective on SDE-based inference. Instead of using a path-based inference formulation over the whole trajectory, it takes a marginal-based view and performs inference directly on the joint distribution at the observed timestamps, while still ensuring that the posterior is induced by a valid underlying SDE. This helps avoid many of the issues in existing methods, such as instability, high computational cost, and limited expressiveness.

**Compliance With Llm Reviewing Policy:**

Affirmed.

**Final Justification:**

Solve my all questions and concerns.

**Key Questions For Authors:**

NA

**Strengths And Weaknesses:**

Strength:
1. The paper is generally very well written and easy to follow.
2.  I think the main angle is quite novel. Instead of using a path-based inference formulation over the whole trajectory, the paper takes a marginal-based view and performs inference at the observation timestamps.
3. The theory is clear and well structured, and it does not feel like something that was artificially inserted into the paper.

Weakness:
1. A main selling point of SDE-based latent-variable models is uncertainty modeling, yet the experiments primarily report point-prediction metrics such as accuracy and MSE. This leaves it unclear whether the learned posterior is actually well calibrated. Please include uncertainty-focused evaluation: ECE/Brier/NLL for classification, and predictive NLL/CRPS/interval coverage calibration for regression, interpolation, and extrapolation.
2. The paper does mention efficiency in the main text, but the actual efficiency experiment is only in the appendix. I think it would be better to include at least one direct efficiency result in the main paper.

---

> ### Author Rebuttal · Authors · 2026-03-31
>
> We genuinely appreciate your review and constructive feedback. We are encouraged by your recognition of the paper's clarity, the novelty and integrity. We will respond to the weaknesses you raised point by point below.
>
> **Q1:  A main selling point of SDE-based latent-variable models is uncertainty modeling, yet the experiments primarily report point-prediction metrics such as accuracy and MSE. This leaves it unclear whether the learned posterior is actually well calibrated. Please include uncertainty-focused evaluation: ECE/Brier/NLL for classification, and predictive NLL/CRPS/interval coverage calibration for regression, interpolation, and extrapolation.**
>
> Our evaluation protocol follows the established benchmarks in prior literature [1-4]. Because these works only report point-prediction metrics, we adopted the same setup to ensure a direct and fair comparison.
>
> We agree with you that uncertainty-focused evaluation is important for generative modeling. Due to the time constraints of the rebuttal period, we provide a preliminary uncertainty-focused evaluation (Test Gaussian NLL) in Table 1 below, comparing our model against representative baselines on all tasks. We will include the complete uncertain evaluation in the final version.
>
> ---
>
> Table 1: Test Gaussian NLL (mean ± std) on all tasks.
>
> | **Model**         | Activity        | Pendulum         | **USHCN Intra.** | **USHCN Extra.** | **PhysioNet Intra.** | **PhysioNet Extra.** |
> | ----------------- | --------------- | ---------------- | ---------------- | ---------------- | -------------------- | -------------------- |
> | CRU               | 0.53 ± 0.04     | -2.14 ± 0.05     | **-5.34 ± 0.01** | -4.15 ± 0.08     | -4.25 ± 0.06         | -3.96 ± 0.05         |
> | NCDSSM            | 0.65 ± 0.07     | -2.45 ± 0.03     | -5.20  ± 0.03    | -4.07 ± 0.08     | -4.45 ± 0.07         | -4.06 ± 0.04         |
> | ACSSM             | 0.67 ± 0.02     | -2.48 ± 0.02     | -5.32  ± 0.01    | -4.52 ± 0.0      | -4.42 ± 0.04         | -3.96 ± 0.08         |
> | **SDE-VI (Ours)** | **0.28 ± 0.05** | **-2.55 ± 0.03** | -5.31 ± 0.02     | **-5.10 ± 0.15** | **-4.60 ± 0.06**     | **-4.14 ± 0.04**     |
>
> [1] Multi-Time Attention Networks for Irregularly Sampled Time Series
> [2] Modeling Irregular Time Series with Continuous Recurrent Units
> [3] Neural Continuous-Discrete State Space Models for Irregularly-Sampled Time Series
> [4] Amortized Control of Continuous State Space Feynman-Kac Model for Irregular Time Series
>
> **Q2: The paper does mention efficiency in the main text, but the actual efficiency experiment is only in the appendix. I think it would be better to include at least one direct efficiency result in the main paper.**
>
> Thanks for your suggestion. We will relocate the efficiency experiment from the appendix to the main text in our final version.

---

> > ### Author Rebuttal · Reviewer_8Ph7 · 2026-04-02
> >
> > Solved and I will increase my score.

---

### Decision · Program_Chairs · 2026-04-30

**Decision:**

Accept (spotlight)

**Comment:**

The reviewers agree that this is a novel, interesting and useful work. There were some concerns about the presentation, theoretical and empirical results, but these were convincingly addressed by the authors in their rebuttal. Please go over the reviews and your rebuttals while revising the manuscript.

PS: Please also go over your references carefully. I see two issues: 1) Citations from the supplementary material (that are not in the main text) seem to be present in the main reference list, and 2) at least 1 citation (for Adam) is a bit botched up.